# Pathways at the Iberian crossroads: Dynamic modeling of the middle–upper paleolithic transition

Yaping Shao[1]*, Konstantin Klein[1], Christian Wegener[1], Gerd-Christian Weniger[2]

**1** Institute for Geophysics and Meteorology, University of Cologne, Cologne, Germany, **2** Institute of Prehistory, University of Cologne, Cologne, Germany

\* yshao@uni-koeln.de

## Abstract

Archaeological and genomic data provide essential yet static insights into human expansion, offering limited understanding of the underlying dynamic processes. As a complementary alternative, we present a high-resolution model of population dynamics and apply it to reconstructing the Middle-to-Upper Paleolithic Transition (MUPT) in Iberia. Through ensemble simulation, we examine Neanderthal (NEA) persistence, modern human (AMH) arrival, and possible interbreeding. The model maps population networks, mobility, and interactions, showing NEAs were confined mainly in coastal refugia and already declining when AMHs arrived. Heinrich Event 5 likely accelerated NEA extinction through climate stress and demographic collapse. AMHs expanded rapidly into Cantabria, overlapping with NEAs and allowing for possibly 2–6% admixture. New dispersal corridors are predicted, showing AMHs moved along the Atlantic coast from southern France into Cantabria, then inland via the Duero Route into Portugal and central Mesetas. By linking climate, demography, and culture, our dynamic model offers a broader explanatory framework that enhances the interpretive power of archaeological and genomic records.

## Introduction

The Middle to Upper Paleolithic Transition (MUPT) between Heinrich Event 5 and 4 (HE5–HE4), approximately 50–38 kyr cal BP (ka hereafter), is marked by the decline and extinction of Neanderthals (NEAs), the emergence and expansion of Anatomically Modern Humans (AMHs), the disappearance of Middle Paleolithic technologies, and the rise of Upper Paleolithic technologies. This transition unfolded against a backdrop of significant climate change (see Fig C in S1 Appendix). The Iberian Peninsula has long been a focal point for MUPT studies, particularly due to ongoing debates about the possible prolonged survival of NEAs in the region [1–11]. Seminal archaeological contributions have shaped this field [12–14]. Research into NEA–AMH admixture and gene flow has been a central focus. Recent findings suggest that most NEA gene flow into AMHs likely occurred during a single period between 50.5–43.5

**Data availability statement:** All Data and Code is available in this external Repository if not otherwise cited within the manuscript from the figshare database (accession number: 10.6084/m9.figshare.30529043).

**Funding:** This study is funded by the Deutsche Forschungsgemeinschaft (DFG, German Research Foundation) via the Collaborative Research Centre 806 (CRC 806, Project ID 57444011) and the Ministry for Culture and Science of North Rhine-Westphalia of Germany (Profile Building 2022 PB22-081).

**Competing interests:** No competing interests.

ka [15] or 49.0–45.0 ka [16]. These studies attempt to reconstruct an extraordinarily complex process from sparse and uncertain data. A comprehensive review by [17] critiques persistent misinterpretations of archaeological evidence and challenges the outdated Neanderthal–modern human dichotomy. Instead, it advocates for an interpretive framework grounded in communities, populations, and short-term historical processes.

The exact timing of NEA extinction in Iberia remains uncertain [7,8,18–20]. NEA fossils from El Sidrón have been dated to 48.4 $\pm$ 3.2 ka [21]. Fragmentary remains from Sima de las Palomas de Cabezo Gordo, found alongside burned faunal bones, have yielded dates of 42.01 and 38.4 ka [5,18], though the reliability of these dates has been questioned [7]. While direct NEA fossils are rare in Iberia, Middle Paleolithic assemblages—commonly associated with NEA presence—are abundant [22]. Several of these sites yield dates younger than 45 ka, supporting the hypothesis of prolonged NEA survival [18]. However, recent re-evaluations have revised some previously assumed Late MP layers to older dates, as seen at Gruta da Oliveira [10]. To date, approximately 100 MP sites have been documented on the Iberian Peninsula. Of these, a dozen — pending further validation — are likely younger than 42 ka in mean age [23]. The remaining sites suggest an earlier abandonment by NEAs, likely before 45 ka [21], pointing to a population decline during HE5 and their final disappearance before or during HE4.

However, several sites featuring Châtelperronian (Chât) technology, dated to around 40 ka [24], suggest a possible temporal and spatial overlap between NEA and AMH populations in Iberia [25,26]. Genomic and anthropological evidence confirms interbreeding between NEAs and AMHs in regions such as southwestern Asia and eastern Europe [15,16,27–29], but whether such interactions occurred on the Iberian Peninsula remains uncertain.

The first AMHs associated with the Aurignacian (AUR) culture are believed to have reached Cantabria from eastern Europe around 42±1 ka [25,30,31]. However, opinions diverge on the subsequent AMH dispersal across the Iberian Peninsula. Zilhão's "Ebro frontier" model suggests that southern Iberia served as the final stronghold of NEAs [1,14]. In contrast, Haws et al. [13] report evidence of AMH occupation at Lapa do Picareiro in Portugal during the first Aurignacian settlement phase (AUR-P1), between 41.1 and 38.1 ka, indicating a more rapid and extensive spread of AMHs than previously assumed — a conclusion disputed by Zilhão and colleagues [14,17]. If validated, such findings would necessitate a revision of prevailing models of the MUPT. However, the apparent isolation of Lapa do Picareiro from other AUR-P1 sites poses a challenging question: how did AMHs reach this location in the first place? Addressing such significant yet contentious issues requires dynamic modelling of AMH dispersal and the underlying population networks.

The MUPT coincided with pronounced millennial-scale climatic fluctuations, as evidenced by a range of paleoclimatic proxies: peat bogs [32], speleothems [33,34], marine sediments [35–38], lake sediments [39], pollen records [40], and loess stratigraphy [20]. These records document cyclical shifts aligned with the Greenland Interstadial (GI) and Stadial (GS) phases. GS periods were characterized by much colder and drier conditions across Iberia. The arrival of anatomically modern humans

(AMHs) in northern Iberia around 42±1 ka was soon followed by the onset of the GS10–GS9/HE4 cold phase, lasting approximately 3,000 years. While AMH populations appear to have retreated across much of Europe during this interval, the extent to which these climatic perturbations affected AMH dispersal in Iberia — and thereby influenced the course of the MUPT — remains an open question.

In summary, three major issues concerning the MUPT remain unresolved: (1) What was the demographic and spatial status of NEAs during this transition? (2) How did AMHs expand across the Iberian Peninsula, and did they interact with NEAs? (3) To what extent did climate change influence the MUPT? To address these questions, we employ a high-resolution, forward-time population dynamics framework — the Our Way – Constrained Agent-Based Model (OW-CABM). This incorporates several innovative features, most notably the Human Existence Potential (HEP), a unified metric that synthesizes climatic, environmental, and archaeological data while accounting for population pressure. HEP functions as the principal driver of both demographic growth and spatial dispersal. In the OW-CABM, we simulate the expansion, interaction, and distribution of NEA, AMH, and MIX (NEA–AMH admixture) populations, each represented as a set of agents. Although not the main focus of this paper, the evolution of population distribution patterns allows for the analysis of network formation, including population fluxes, connectivity, stable settlement centers, and least-cost paths. The agents follow probabilistic rules at the microscopic level, giving rise to emergent macroscopic patterns. Unlike conventional agent-based models, OW-CABM constrains agent behavior through macroscopic HEP values. Due to their distinct cultural and climatic adaptations, NEAs and AMHs exhibit different HEP profiles, but interbreeding is possible in zones of demographic overlap. The model operates in ensemble mode, enabling robust statistical evaluation of outcomes and associated uncertainties. In addition to addressing the three core questions of the MUPT, OW-CABM provides new insights into long-standing debates — such as the identification of NEA refugia and the dispersal pathways of AMHs. By capturing the dynamic interplay of climate, culture, and demography, our simulations offer a level of explanatory depth that cannot be achieved through archaeological or genomic data alone.

## Model, data and numerical experiments

### Framework of constraint agent-based model

Human dispersal models have been under development since the 1960s [41–44]. For continent scale problems, diffusion/reaction models have been used [45–47]. Models coupling large climate and archaeological data sets have been developed to study climate impacts on human dispersal [23]. Agent-based models constructed in recent years [48] are advantageous in modeling human dispersal and admixture based on microscopic understanding of population dynamics. We present here a new constrained agent-based model.

We model human population distributions as the manifestation of the human system which has biological, cultural and environmental dimensions. The processes on these dimensions interact, causing complex feedbacks and human dispersal. A framework is proposed here to incorporate the three dimensions for quantitative modeling of human dispersal. It consists of the components of climate-environment and archaeological data, HEP model and human mobility model. We use HEP as a unifying quantity which drives the population dynamics. HEP has three layers of information, namely, the climate-environment HEP $\Phi_E$, accessible HEP $\Phi_{Ac}$, and available HEP $\Phi_{Av}$.

The climate-environment HEP $\Phi_E$ measures whether climate-environment conditions are suitable for human biological existence and whether resources are abundant. The input for computing $\Phi_E$ are either primary (e.g., temperature and rainfall) or secondary variables (e.g., net primary production) or both. To determine the preferential climate-environment conditions for human existence of a given culture, knowledge a priori and/or information derived from archaeological data is required. The quantity $\Phi_E$ sets the upper limit for human existence, but this limit is very loose because human existence is also limited by the human capacity to harness resources and adapt to environmental changes.

The accessible $\Phi_{Ac}$ is an extension of the concept of cultural carrying capacity used in studies of population dynamics [49,50]. In our study, $\Phi_{Ac}$ is not a constant but a function of space and time, depending both on climate-environment

conditions and human activities. $\Phi_{Ac}$ describes the overarching effect of the cultural dimension of the human system, and embedded in it are processes such as technological progress, societal structure, cultural behavior, religion etc. The estimation of $\Phi_{Ac}$ is not straight forward, but relies on the parameterization of the anthropological processes and the use of machine-learning from archaeological data [51].

Regions of high $\Phi_{Ac}$ are attractive to humans, where the population is likely to grow to a critical level. Beyond this level, the availability of resources per capital becomes a limiting factor which drives humans to other more favorable areas or to reduce the reproduction rate. To reflect this understanding, the available HEP $\Phi_{Av}$ is introduced. This is both a driver of the population dynamics and a dependent on population density, highlighting the non-linear dynamics of the human system. The framework of the OW-CABM is as illustrated in Fig 1.

## Archaeological data

For the HEP calculation for the NEAs, 99 MP sites in Iberia are used (Fig A, Table C in S1 Appendix, [22]). Most of the sites are dated and assigned to the Mousterian techno-complex. Seven of these sites show evidence of another techno-complex in a sublayer of the stratigraphy, the Chât which represents a transition phase from the Middle Paleolithic to Upper Paleolithic. Human fossils found in the Chât layers regularly belong to NEAs. Therefore, we assumed that NEAs were the makers of the Chât. All MP sites are included for training the HEP model for the NEA population, although some of the sites might no longer be inhabited at the time of GI11-10.

The inclusion of Chât sites as evidence of NEA presence remains debated due to uncertain authorship [24,52], a issue that recent studies have not resolved [53–55]. In Iberia, the Chât seems to be linked to population oscillations among final NEA groups that align with HE5 and HE4 events, though these dynamics are too fine-scaled for current chronometric methods. In our study, we assigned the few northern Iberian Chât sites to late NEAs (Table C in S1 Appendix), but their inclusion did not alter the HEP patterns of the NEAs (cf. Fig A and Fig I in S1 Appendix) and our core results. For more discussions, see S1 Appendix.

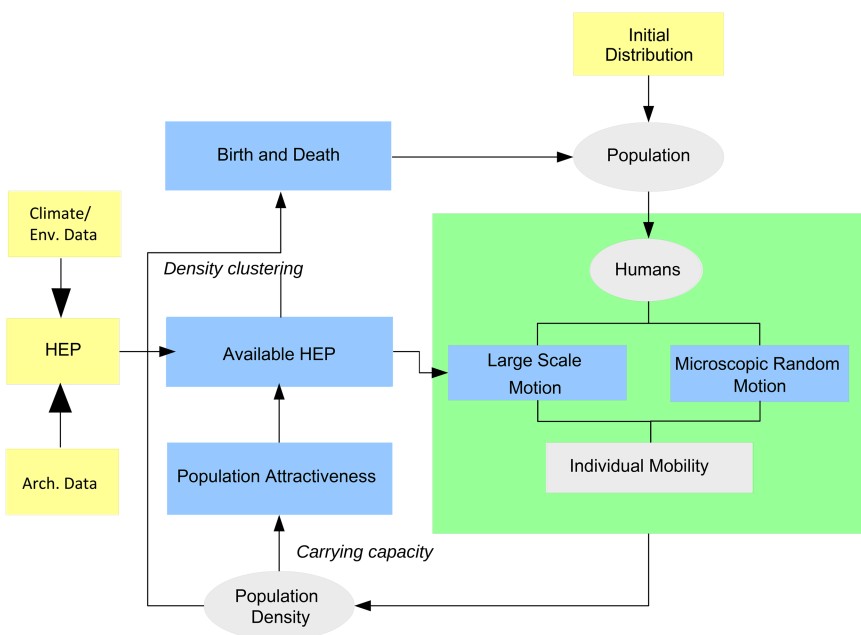

**Fig 1**. **Schematic illustration of the OW-CABM scheme.**

For training the HEP model for the AMH population, the Aurignacian (AUR) sites are used. The AUR is divided into two phases, AUR-P1 and AUR-P2, which correspond approximately to the periods 43-37 ka and 37-32 ka, respectively. Here, only the 66 AUR-P1 sites located to the south of 47°N and west of 6°E are used. For the AUR-P1, as listed in Table D in S1 Appendix [56]. Two sites required special attention, the Lapa do Picareiro site and the Cueva de Bajondillo site. For both, the respective excavators postulate their classification in the AUR-P1 [13,57]. Due to uncertainties associated with the Cueva de Bajondillo site [58,59], it is excluded for the HEP training. In the case of Lapa do Picareiro, the small stone tool inventory of less than 50 pieces can be assigned to an Aurignacian. However, the early radiocarbon dates come from the area below or from the lower edge of the find layer. Therefore, it cannot be decided with certainty that they actually date the find context. The Lapa do Picareiro site is placed as a test site in the analysis.

## Climate data

The generation of the paleoclimate used here is the same as described by [11]. The COSMOS [60] is used to simulate the GI11-10 and GS10-9/HE4 global climate conditions. For each of the simulations, a data set covering 35 model years is created after a spin up of 1400 model years. For both simulations, the boundary conditions are specified to those at 40 ka [61]. The Earth's orbital parameters and concentrations of trace gases are set correspondingly. To model the GS10-9/HE4 conditions, fresh water release to the North Atlantic [62] is imposed. The GCM simulated atmosphere climate data has a spatial resolution of ~ 3.75° with 19 vertical layers. The ocean model, including a thermodynamic sea ice model, employs a bipolar curvilinear model grid with a formal horizontal resolution of 3.0° x 1.8° and 40 unevenly spaced vertical layers.

To increase the climate-data resolution for HEP modelling, regional climate simulations are carried out by nesting the Weather Research and Forecasting (WRF) model [63] into the COSMOS runs. The WRF runs for GI11-10 and GS10-9/HE4 last each for 35 years in multiple nesting to a resolution of approximately 12.5 km.

The land ice cover data for the WRF runs are taken from reconstructions using the ICE-6G-C model of the PMIP4 database [64]. For the GI11-10 run, the ice cover is assumed to arise from a sea level of -72 m (relative to today's sea level), and for the GS10-9/HE4 run to arise from a sea level of -96 m. The vegetation is set as [65]. The topography over the ice sheet as well as the coastline are also adapted based on the respective ICE-6G-C reconstructions used for GI11-10 and GS10-9/HE4. The same orbital parameters and trace gas concentrations as set in the COSMOS model are used in the WRF model [66,67].

## Human existence potential

The Human Existence Potential (HEP) is computed using the logistic regression methodology of [11], which integrates paleoclimate data and archaeological records. The model combines a second-order polynomial of climate variables with modification functions that account for resource accessibility.

Paleoclimate data are reconstructed on a two-dimensional grid with a resolution of 20 × 20 km, somewhat coarser than the native 12.5 km WRF model output [68]. To mitigate site-clustering bias, archaeological data are aggregated by grid cell, equivalent to the spatial-blocking technique [69], where any cell containing one or more sites is treated as a single presence record. Ocean cells are excluded.

We classify cells containing one or more archaeological sites as presence cells (HEP = 1). The remaining terrestrial cells are designated as absence cells which are subdivided into two categories: a-priori absence and pseudo-absence.

A-priori absence cells (HEP = 0) represent regions where human presence was considered impossible based on pre-defined bioclimatic thresholds. For Anatomically Modern Humans (AMHs), these criteria are: (a) an annual mean temperature (BIO1) below $-2\,°C$ or above $16\,°C$, and (b) precipitation in the wettest month (BIO13) below 30 mm or above 250 mm. These thresholds correspond to the 95% confidence limits derived from the probability density functions of these

variables at known archaeological sites. The incorporation of this expert knowledge is a critical step to prevent physiologically implausible outcomes that can arise from purely data-driven models and is a necessary prerequisite for the logistic regression, which requires a defined set of true absences.

Pseudo-absence cells represent areas of uncertainty and may correspond to one of three scenarios [70]: (1) methodological absences (human presence occurred but no record was preserved or discovered), (2) contingent absences (areas were potentially habitable but remain unexplored), or (3) environmental absences (areas were unsuitable for habitation).

Given the ambiguous nature of these cells, we evaluated three treatments for model training: (1) Excluding pseudo-absence cells entirely and training solely on presence and a-priori absence records; (2) Assuming equal probability for the three absence types, using one-third of pseudo-absence cells as absences and one-third as presences; and (3) Weighting the probability of a cell being a presence based on the observed discovery rate in the dataset, defined as $N_p/(N_p + N_{pa})$, where $N_p$ and $N_{pa}$ are the number of presence and pseudo-absence records, respectively.

Sensitivity tests indicated that the differences between these options are minor. We adopted Option (3) for this study, as it is the most stringent approach and results in slightly more conservative (lower) HEP estimates. While this treatment introduces some uncertainty, the HEP model remains robust, as it is primarily constrained by the well-defined presence and a-priori absence cells.

From the resulting presence/absence records, a multivariate second-order logistic regression model is constructed. This approach was chosen for its computational efficiency, allowing HEP estimation at any spatiotemporal coordinate $(x,y,t)$ where predictor values are known.

Seventeen bioclimatic variables [71] were computed from WRF data (Table 1) and standardized. Predictor selection prioritized low collinearity, assessed via hierarchical correlation clustering [51], and strong discriminatory power. The final predictors—Bio1, Bio4, Bio16, Bio15, and Bio17—were used for both study periods.

A quadratic polynomial $q$ is constructed from the selected predictor vector $\vec{P}$:

$$q(\mathbf{P}) = \frac{1}{2}\mathbf{P}^T\underline{A}\mathbf{P} + \mathbf{B} \cdot \mathbf{P} + C \tag{1}$$

**Table 1**. Definition and clusters of the 17 bioclimatic variables considered as the candidate predictors of the HEP.

| Cluster | Bioclim Var | Definition |
|---|---|---|
| Mean temp. | Bio1 | Annual Mean Temp. |
| (T-mean) | Bio5 | Max Temp. of Warmest Month |
| | Bio6 | Min Temp. of Coldest Month |
| | Bio10 | Mean Temp. of Warmest Quarter |
| | Bio11 | Mean Temp. of Coldest Quarter |
| Temp. seasonality | Bio4 | Temp. seasonality |
| (T-var) | Bio7 | Temp. Annual Range |
| Daily temp. variation | Bio2 | Mean Diurnal Range |
| | Bio3 | Isothermality |
| Mean precip. | Bio12 | Annual Precip. |
| (P-mean) | Bio13 | Precip. of Wettest Month |
| | Bio16 | Precip. of Wettest Quarter |
| | Bio19 | Precip. of Coldest Quarter |
| Precip. seasonality | Bio14 | Precip. of Driest Month |
| (P-var) | Bio15 | Precip. Seasonality |
| Mean dryness | Bio17 | Precip. of Driest Quarter |
| (D-mean) | Bio18 | Precip. of Warmest Quarter |

where $\underline{A}$, **B**, and $C$ are parameters estimated during model training. The HEP is then mapped via the logistic function:

$$\Phi_E(q) = \frac{1}{1 + \exp(-q)} \tag{2}$$

We trained the model through 1000 ensemble runs, each using a random 80% subset of presence/absence records for training and the remaining 20% for validation. The ensemble mean, denoted $\langle\Phi_E\rangle$, provides the final HEP estimate and shows good performance as demonstrated in [11].

Static environmental factors not captured by climate variables—including water bodies, topography, and forests—are incorporated through modification functions. These factors, which remain relatively constant over cultural timescales but affect different techno-complexes variably, are used to compute the accessible HEP:

$$\Phi_{Ac} = \langle\Phi_E\rangle \cdot g_1 \cdot g_2 \cdot g_3 \cdot g_x \tag{3}$$

where $g_1$ and $g_2$ represent topographic accessibility based on elevation and roughness [72], defined by the piecewise linear function:

$$g_{1\,or\,2}(x) = \begin{cases} 1.0, & x < x_l \\ 1.0 - (x - x_l) \cdot m, & x_l \leq x < x_u \\ 0.8, & x \geq x_u \end{cases} \tag{4}$$

with $m = 0.2/(x_u - x_l)$. Parameter values are population-specific: for AMHs, $g_1$ uses $x_l = 350$ m and $x_u = 2000$ m, while $g_2$ uses $x_l = 70$ m and $x_u = 400$ m. For NEAs, $g_1$ parameters are $x_l = 450$ m and $x_u = 2000$ m; roughness is excluded for MP sites due to unclear patterns. Water bodies ($g_3 = 0$) and dense forests ($g_x = 0$) are considered uninhabitable.

The model's robustness has been extensively validated [11,31,51,73]. Ensemble analysis reveals generally low uncertainty, though higher variances occur in peripheral regions like central Iberia, where interpretations require caution. While HEP patterns are largely stable, estimates remain sensitive to archaeological data limitations and paleoclimate uncertainties. Outlier sites such as Lapa do Picareiro can induce localized shifts in HEP with demographic implications.

Our high-resolution (12.5 km) Iberian HEP estimates reveal finer spatial patterns than pan-European models [31,74], which used broader spatiotemporal data at coarser resolution (50 km). This methodological focus enables detection of subtle demographic connections but may yield quantitative differences from large-scale approaches.

## Population growth

Human population density ($\rho$) is number of humans per unit area, often expressed in "number of humans/100 km$^2$" which we call PDU (population density unit). The conservation equation for $\rho$ is

$$\frac{d\rho}{dt} = p \tag{5}$$

where $t$ is time and $p$ is population growth rate. Both $\rho$ and $p$ are functions of space and time. We can rewrite Eq (5) as

$$\frac{\partial\rho}{\partial t} + \boldsymbol{v} \cdot \boldsymbol{\nabla}\rho - D\nabla^2\rho = p \tag{6}$$

where $\boldsymbol{v}$ is a drift velocity and $D$ diffusion coefficient. Eq (6) describes human dispersal as a drift, diffusion and production process. The quantities $\boldsymbol{v}$, $D$ and $p$ depend on $\Phi_{Av}$.

Two competing processes influence how $\Phi_{Av}$ is related to $\rho$. First, humans are gregarious and hence regions of low population are less attractive for migration, where mutual support is needed to increase the efficiency in utilizing the accessible resources and reproduction. Second, if the population exceeds a certain level, the resources available per capital become scarce, creating a population pressure. We define $\rho_c$ as

$$\rho_c(x, y, t) = C\Phi_{Ac}(x, y, t) \tag{7}$$

where $C$ is the cultural carrying capacity and $\rho_c$ is the local carrying capacity. In OW-CABM, $C$ is a parameter to be specified. For AMH cultures in the Upper Paleolithic, $C$ is of the order of 1 to 10 PDU and for NEA cultures in the Middle Paleolithic, it may be smaller. It is important to note that we do not assume here a lower technological and cognitive ability of the NEAs than that of the AMHs, and thus in this study, we select similar $C$ values for both species.

The method for the estimation of $\Phi_{Ac}$ follows [31], which is briefly repeated here. The above-mentioned two competing processes are represented using an attractiveness function $f_{pa}$ of the form of a Weibull distribution

$$w = \frac{\eta}{\epsilon} \left( \frac{\rho/\rho_c}{\epsilon} \right)^{(\eta-1)} \exp\left[ \left( \frac{\rho/\rho_c}{\epsilon} \right)^{\eta} \right] \tag{8}$$

where $\epsilon$ is the scaling and $\eta$ the shape parameter. Using Eq (8) we define

$$f_{pa} = w/w_{max} \tag{9}$$

with $w_{max}$ being the maximum of $w$. $\Phi_{Av}$ is then expressed as

$$\Phi_{Av} = f_{pa} \cdot \rho_c \tag{10}$$

As $\Phi_{Ac}$ is scaled from 0 to 1, $\Phi_{Av}$ is scaled from 0 to $C$. Examples of $\Phi_{Av}$ as function of $\rho$ for two different $\rho_c$ values, with parameters $\epsilon$ and $\eta$ in Eq 8 set to 0.4 and 1.6, respectively, are shown in [31].

We follow Verhulst (1838) and rewrite Eq (5) as

$$\frac{d\rho}{dt} = \rho \, r_B \left( 1 - \frac{\rho}{\rho_c} \right). \tag{11}$$

with the term on the r.h.s. being $p$ and $r_B$ the relative population growth rate. Eq (11) shows that if $\rho \to \rho_c$, then $p \to 0$ and for $\rho > \rho_c$, $p$ is negative. For AMHs of the Aurignacian, $r_B$ is the order of 0.01 yr$^{-1}$. Lower or higher values have been used by other researchers, e.g., 0.004 yr$^{-1}$ in [75] and 0.031 yr$^{-1}$ in [76]. Both $r_B$ and $C$ for the NEA and AMH populations are uncertain and must be set as tunable parameters. The OW-CABM simulates the population growth using a module which accounts for the birth or death of the individual humans in each grid cell, but the net population growth rate satisfies Eq (11).

Eq (11) as a semi-empirical population growth model has been widely used, but has limitations. The birth-death processes on the microscopic scale (e.g., individual humans and groups) are highly complex. For example, the need for hunter-gatherer groups to move in an annual territory has an impact on their individual members. Children and elderly and sick humans are particularly vulnerable if the necessity to move over large distances increases. The high mortality rates of these groups may lead to feedbacks which negatively impacts on population resilience to changes. Mating network is hugely important to the survival of humans. The studies of [77–79] suggested that a certain population density is necessary for the mating network to sustain. Clearly, the collapse of the mating network will lead to a zero or negative $r_B$. Under

optimal mating conditions, it is justified to assume $r_B$ to be positive, but as $\rho$ falls below a critical value, such that mating becomes difficult, $r_B$ must reduce to a negative value (we note that the same effect can be achieved by reducing $\Phi_{Av}$ to zero). A simple approach is taken here by setting

$$r_B = r_o \left[ \frac{1-\eta}{1+\exp(-x)} + \eta \right] \tag{12}$$

with

$$x = \epsilon \ln(\rho/\rho_d)$$

and $\eta$ being a small negative value, such that $r_B = \eta r_o$ for $\rho \to 0$. Fig 2 shows an example of $r_B$ as function of $\rho$.

## Human mobility

Eq (6) is solved in the Lagrangian framework by tracking the motion of individual humans (agents). An important feature of the OW-CABM is that human motion is not a pure diffusion process, but composed of a macroscopic drift and a microscopic random walk, i.e., constrained random walk. Different constraints are imposed on the Lagrangian stochastic model for human motion on the macroscopic and microscopic levels.

The macroscale constraint is applied because humans migrate to areas of favorable conditions. For example, the dispersal of AMHs out of Africa is not a linear process, but in waves of propagation linked to climatic phases [80] which provided better conditions for human existence. According to [81], the history of human settlement are adaptive cycles, each with the successive phases of population growth, conservation, distortion and reorganization. This idea of repeated-replacement is reflected in the constraint that human dispersal on the macroscale is dictated by the gradient of $\nabla \Phi_{Av}$.

On microscale, human motion is assumed to be stochastic. We thereby respect that the behavior of the individuals cannot be precisely modeled, but their ensemble statistics must be consistent with the ethnographic understanding. In OW-CABM, the random motion produces population fluctuations around human population centers and diffusion away from them.

Consider agent $n$ in a population of $N$ with position vector $\boldsymbol{X}_n$ and velocity vector $\boldsymbol{U}_n$. Based on the above discussions, the stochastic differential equations can be written as

$$d\boldsymbol{U}_n = (\alpha C \nabla \Phi_{Av} - \gamma \boldsymbol{U}_n)\, dt + \beta d\boldsymbol{W}_n \tag{13}$$
$$d\boldsymbol{X}_n = \boldsymbol{U}_n dt \tag{14}$$

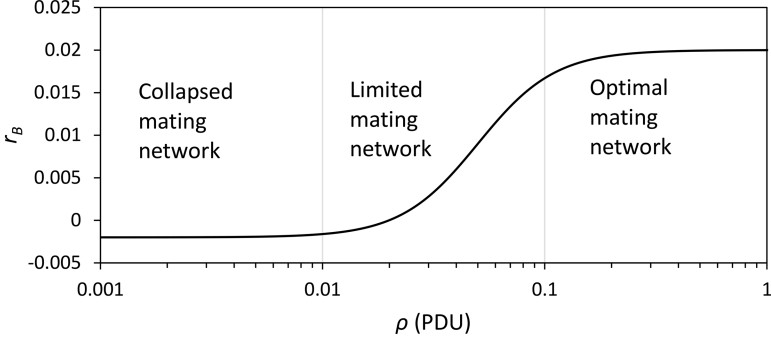

**Fig 2**. Relative population growth rate as function of population density $\rho$ (PDU), by setting in Eq (12) $\rho_d$ = 0.05 PDU, $\epsilon$ = 2.5, $r_o$ = 0.02 and $\eta$ = −0.1.

where $\beta$ is a diffusion coefficient and $d\boldsymbol{W}$ Gaussian white noise. The coefficient $\alpha$ [km$^2$ yr$^{-2}$] relates the acceleration of the macroscopic drift velocity to the gradient of $\Phi_{Av}$ and is modelled with

$$\alpha = \frac{U_s \cdot G_d}{D_t}. \tag{15}$$

where $G_d$, $U_s$ and $D_t$ are scaling length, velocity and time, respectively. Knowing the position $\boldsymbol{X}_n$ and velocity $\boldsymbol{U}_n$, the population density and population flux associated can be computed.

The AMHs took several thousand years to expand from Africa to Europe [80,82]. This gives a scaling drift velocity $U_s$ of the order of 1 to 10 km yr$^{-1}$, similar to that of farming spread in Europe during the Neolithic [83,84]. As humans migrate to new regions, they encounter unfamiliar environments. The response time of humans to changes is represented using $D_t$. For hunter-gatherers, we set $D_t$ to 1 to 10 years. For a smaller $D_t$, humans respond faster to the changes and vice versa. To relate macroscale drift to the gradient of $\Phi_{Av}$, a drift-distance-scale $G_d$ is introduced. Suppose the magnitude of the $\Phi_{Av}$ gradient that corresponds to $U_s$ be $|\boldsymbol{\nabla}\Phi_{Av}|_s$. Then, as $\Phi_{Av}$ varies between 0 and $C$, we have

$$|\boldsymbol{\nabla}\Phi_{Av}|_s = \frac{C}{G_d}.$$

A smaller $G_d$ implies that a steeper gradient is necessary to match $U_s$. For hunter-gatherers, $G_d$ is the order of 500 km. From Eq (15), we found $\alpha$ to be in the range of 500 to 5000 km$^2$ yr$^{-2}$. The $\alpha$ value we used in most of our simulations is 1250 km$^2$ yr$^{-2}$.

The parameters $\beta$ (km yr$^{-3/2}$) and $\gamma$ (yr$^{-1}$) in Eq (13) characterize the human random motion, where

$$\beta = \sqrt{\sigma^2 \cdot \gamma}. \tag{16}$$

is the diffusion coefficient and

$$\gamma = \frac{1}{\tau} \tag{17}$$

is the damping coefficient, with $\tau$ being the Lagrangian time scale and $\sigma$ the standard deviation of velocity fluctuations. The term $-\gamma \boldsymbol{U}_n$ in Eq (13) ensures the motion of a human is damped due to interferences by the motions of other humans. The Lagrangian time scale, $\tau$, is the autocorrelation time scale of human motion under the influences of human interactions. For simplicity, $\tau = D_t$ is assumed.

Embedded in $\beta$ are both human's inspiration for exploration and the local variability of the environment, climate and demographic conditions. In Eq (13), random motion is assumed to be locally isotropic, such that $\beta = |\underline{\boldsymbol{\beta}}|$, with $\underline{\boldsymbol{\beta}}$ being a dispersion matrix, is a scalar. There is no mathematical difficulties to assume the random motion to be locally non-isotropic. In most of our experiments, $\sigma$ is set to 15 km yr$^{-1}$.

## Initial and boundary conditions

For model initialization, $N_0$ humans are Gaussian distributed in population centers with prespecified mean positions and standard deviations. The initial velocities of the humans are assumed to be random of magnitude $\sigma$. Humans outside of the study domain are excluded from consideration.

The boundary of the domain is symmetrically reflective. Water bodies are natural boundaries for human expansion. The $\Phi_{Av}$ values for the sea grid cells are set to zero and consequently, humans approaching the water bodies are systematically driven back. Further, coastlines are treated as symmetric-reflective boundaries.

## Model parameters

The model parameters are listed in Table 2, together with the values specified in the various numerical experiments conducted in this study. For simplicity, the Lagrangian time scale ($\tau$) and the shape parameters of the population attractiveness function ($\epsilon$ and $\eta$) are fixed empirically as such that the model simulations deliver reasonable outcomes from archaeological perspectives.

Building on earlier HEP analyses [11], our sensitivity tests (see S1 Appendix, Table A) show that model parameters quantitatively influence human dispersal in Iberia. Macroscopic drift ($\alpha$) affects dispersal patterns, with smaller values producing wider dispersal and larger values (Fig E in S1 Appendix) promoting dense centers. Similarly, diffusion coefficient ($\sigma_u$) modifies settlement extent: smaller values limit coverage while larger values (Fig D in S1 Appendix) expand it to western Cantabria and central Iberia, and redistribute population centers. Growth rate ($r_B$) controls population size and distribution, from constrained expansion at lower rates to larger populations at higher rates, with complex center replacement at intermediate values (Fig F in S1 Appendix). Cultural carrying capacity ($C$) similarly influences population size, enabling settlement in lower HEP regions at higher values while maintaining distribution patterns across tested values (Fig G in S1 Appendix). These results demonstrate the intricate parameter interplay shaping human settlement.

## Results and discussions

Considering radiocarbon chronology, the MUPT on Iberian Peninsula most likely took place in a weakly-interactive mode, but site stratigraphy so far has not displayed inter-stratification. If NEAs were extinct before the arrival of AMHs, then the MUPT was sequential and no interactions between the two populations were possible. But in the weakly-interactive mode, a small NEA population still existed when the AMHs arrived in northern Iberia and admixture between the two populations might have occurred in some areas. This case is the most interesting, which we divide into three phases:

- Phase I ($\sim$50-43 ka): cold climate, comprising HE5 and GS12, interrupted shortly by G12 (Fig S6), when only NEAs existed;
- Phase II ($\sim$43-41 ka): warm climate, comprising GI11 and GI10, interrupted shortly by GS11, when NEAs existed on the peninsula as AMHs reached northern Iberia, and admixture was possible;
- Phase III ($\sim$41-38 ka): cold climate, comprising GS10 and GS9/HE4, interrupted shortly by GI9. The climate shift from GI11-10 to GS10-GS9/HE4 could have severely impacted on the MUPT.

## Chance of neanderthal survival

What chances did NEAs have to survive the HE5 and possibly HE4? Studies on the subject are many, e.g., [85–89]. One view is "early extinction", i.e., the NEAs became extinct by or soon after the HE5, and the other is "later survival", i.e., they survived until HE4 and even beyond in refuges. Embedded in both these views is that climate change played a decisive

Table 2. **Model parameters and their values used in the main numerical experiments.** The parameter $\alpha$ is a scaling acceleration; $\sigma_u$ standard deviation of random motion velocity; $C$ cultural carrying capacity; and $r_B$ population growth parameter.

| Experiment | $\alpha$ | $\sigma_u$ | $C$ | $r_B$ |
| --- | --- | --- | --- | --- |
| | (km/yr)² | (km/yr) | (PDU) | (1/yr) |
| ExpNEA-C | 1250 | 15 | 2 | 0.02 |
| ExpNEA-H | 1250 | 15 | 3 | 0.05 |
| ExpNEA-L | 1250 | 15 | 1 | 0.01 |
| ExpAUR-C | 1250 | 15 | 3 | 0.02 |
| ExpMIX-C | 1250 | 15 | 2 | 0.02 |

role in reducing the NEA population size [90]. But population size is only one of the many tangled quantities which influence population stability, subject both to external forcing and internal population dynamics as for example reflected in cultural carrying capacity $C$, population growth rate $r_B$, and human mobility.

The OW-CABM described in the Model Section is used with the model parameters listed in Table 2 to investigate the chance of NEA survival. We conducted experiments for the NEA population by setting $C$ between 1 and 3 PDU and $r_B$ between 0.01 and 0.05 yr$^{-1}$. For initialization, 1000 humans are Gaussian distributed with a standard deviation of 1° at four locations at (3.93°W, 42.3°N), (0.72°W, 40.03°N), (5.54°W, 36.66°N) and (8.6°W, 39.2°N). For each experiment, an ensemble of simulations of 1000 members is generated by perturbing all input parameters by 10%. All simulations for the NEA population cover the period of 50 - 38 ka.

Sensitivity tests demonstrate that variations in initial population size, distribution, and location only affect model outcomes during an initial spin-up period of 500–1000 years (Fig H, Table B, S1 Appendix). After this transient phase, all experiments converge, confirming the long-term dynamics are robust to initial conditions. Consequently, the first 300 years of all simulations are excluded from the analysis to eliminate this spin-up effect.

Fig 3 shows the ensemble- and time-averaged NEA population density $\bar{\rho}_{nea}$ for three runs ExpNEA-H, ExpNEA-C and ExpNEA-L, with the values of ($C$, $r_B$) of (3 PDU, 0.05 yr$^{-1}$), (2, 0.02) and (1, 0.01), respectively. These runs represent the cases of high, moderate, and low NEA populations. As seen, $\bar{\rho}_{nea}$ depends strongly on $C$ and $r_B$. For ExpNEA-H, the NEA population was distributed in the areas of Mediterranean, Cantabrian and Portuguese coasts, central upper Meseta and southern tip, all areas of high HEP values (Fig A in S1 Appendix). For ExpNEA-C, similar population distributions as ExpNEA-H are found, but the population centers were weaker and settlement areas smaller. For ExpNEA-L, NEAs existed only in isolated coastal spots, while the interior of Iberia was unpopulated.

The time series of the ensemble mean NEA population size $P_{nea}$ and $P_{nea} \pm \sigma_P$, with $\sigma_P$ being the ensemble standard deviation, are shown in Fig 4.

As mentioned earlier, the long-term behavior of $P_{nea}$ is not too sensitive to its initial value, as $P_{nea}$ starts to follow the inherent dynamics after a model spin-up of less than 500 years (first 300 years not shown). We found (1) that climate change strongly impacted on $P_{nea}$, e.g., for ExpNEA-H it nearly halved in GS9/HE4 compared to that in the earlier warm climate. In GS times, $P_{nea}$ for ExpNEA-H and ExpNEA-C was at low levels; (2) For fixed $C$ and $r_B$, if the NEAs survived the HE5, then they would also survive HE4; and (3) For good $C$ and $r_B$ values, e.g., ExpNEA-H and ExpNEA-C, the HE5

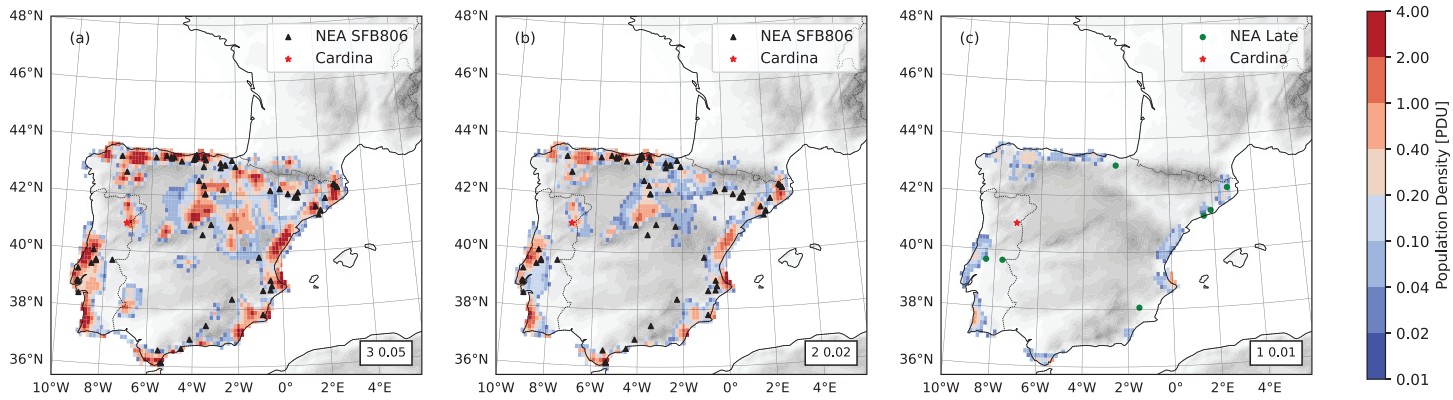

**Fig 3. NEA population density under three different scenarios.** (a) For ExpNEA-H with $C = 3$ PDU and $r_B = 0.05$ yr$^{-1}$, the ensemble- and time-averaged population density $\bar{\rho}_{nea}$ in PDU plotted together with topography (grey shaded) and the MP sites (black triangles); (b) As (a), but for ExpNEA-C with $C = 2$ PDU and $r_B = 0.02$ yr$^{-1}$; and (c) As (a), but for ExpNEA-L with $C = 1$ PDU and $r_B = 0.01$ yr$^{-1}$. Green dots are the youngest MP/CHAT sites to our best knowledge. The red start marks the Cardina site. The site was not used for training the HEP, but the OW-CABM correctly predicted NEA existence here in ExpNEA-H and ExpNEA-C.

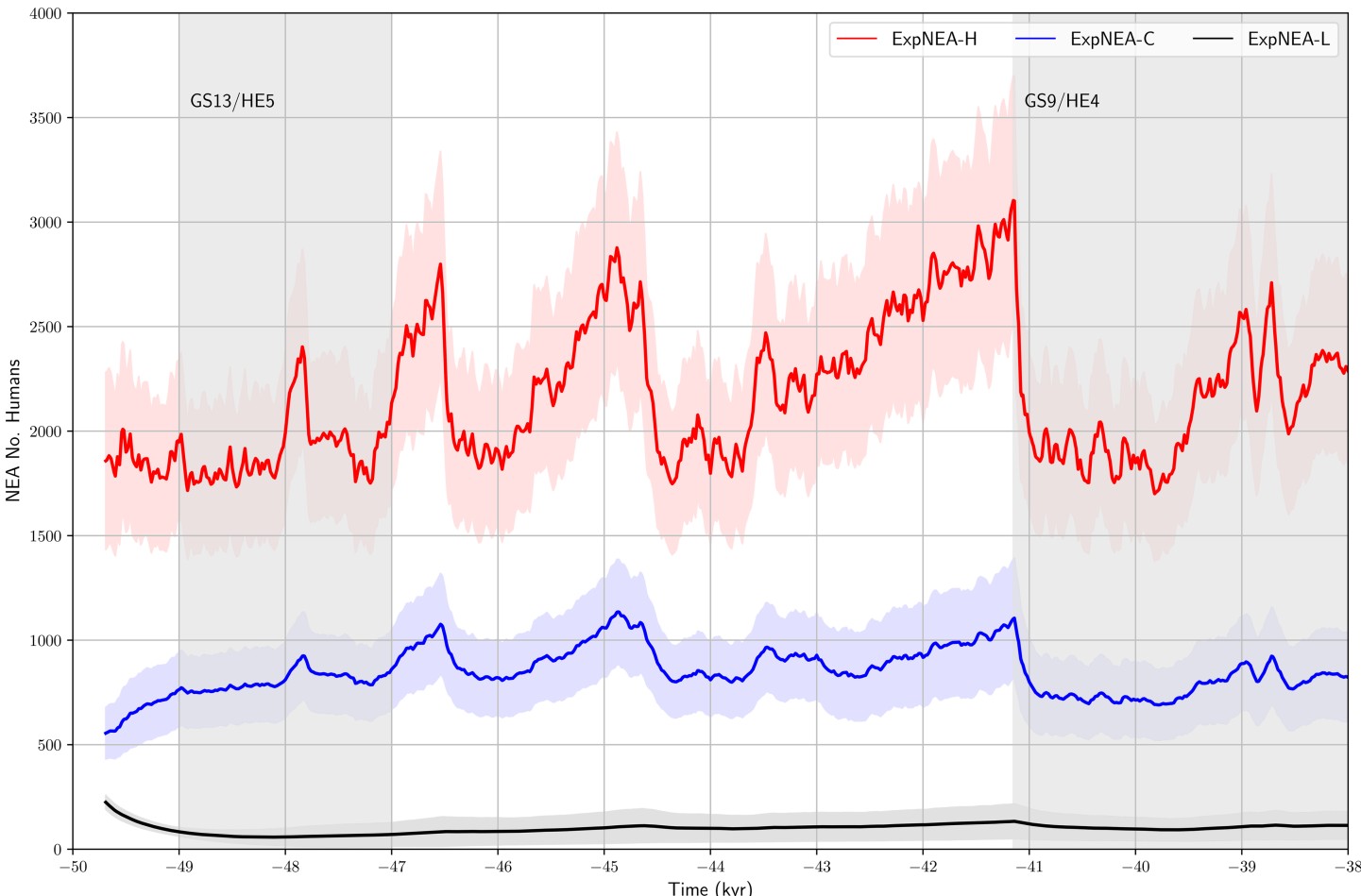

**Fig 4. Ensemble-mean time series of the total NEA population size over the period 50 - 38 ka.** The results are from ExpNEA-H, ExpNEA-C and ExpNEA-L. The thick line represents the ensemble-mean $P_{nea}$ the shaded area $P_{nea} \pm \sigma_P$ with $\sigma_P$ being the ensemble standard deviation.

and HE4 climate change would not be enough to result in NEA extinction. For ExpNEA-L, with very low values of $C$ and $r_B$, a low population of about 100 humans could survive in coastal refuges (Fig 3) over a long period of time, because the HEP values there were relatively stable despite large-scale climate changes. However, the decreases in $C$ and $r_B$ combined with the cold climate shift substantially increase the risks of NEA extinction: $P_{nea}$ reduced to zero in 10%, below 30 in 40%, and below 50 in 60% of the ExpNEA-L runs, mostly during HE5.

Both $C$ and $r_B$ strongly influence the resilience of the population to climate change. Since there is little independent information on the NEA population size, it is difficult to say which of the runs is the most realistic. The agreement of the population density and the distribution of the MP sites is the best for ExpNEA-H (Fig 3a). This implies that NEAs used to have reasonably large $C$ and $r_B$ values, with a $P_{nea}$ of the order of 2000 to 3000.

Genetic and demographic studies yielded various estimates for the NEA metapulation size, which we scale to Iberia assuming it constituted 5% in area of the total range. [91] derived a female effective size that translates to a pan-European census size ($N_c$) of 30000–54000, implying an Iberian population of 1500–2700. [90] found a broader $N_c$ range of 5000–70000, leading to a highly uncertain estimate of 250–3500 individuals for Iberia. In contrast, our interpretation of the PSMC (Pairwise Sequentially Markovian Coalescent) model result in [92] indicates a much larger late-stage $N_c$ of

75000–150000, pointing to an Iberian population of 3750–7500. Clearly, the extrapolation of total pan-European census size to Iberia is subject to profound uncertainties, stemming from the wide disparities in the original metapopulation estimates and the critical yet unverified assumption of a uniform population density across NEA territory, resulting in Iberian estimates that vary by an order of magnitude, from a few hundred to many thousand individuals.

The internal logic of the OW-CABM suggests that the dense settlement and large population size simulated in ExpNEA-H are likely inconsistent with the archaeological record and an improbable state for NEAs during the MUPT. A population with such high cultural capacity $C$ and birth rate $r_B$ would have been sufficiently resilient for NEAs to survive unless they experienced a sudden, unforeseen collapse. The fact that NEAs became extinct suggest that the values of $C$ and $r_B$ were smaller during the MUPT, likely to be between those for ExpNEA-C and ExpNEA-L. With $C>2$ PDU and $r_B > 0.02$ yr$^{-1}$, the NEAs could have survived the HE5 and existed until HE4, while with smaller $C$ and $r_B$ values, the NEAs could only have survived in the refuges but not maintained a large population on the peninsula during the MUPT. We thus argue that ExpNEA-C and ExpNEA-L are likely to be closer to reality, which set the upper and lower limit of the NEA population during the MUPT in weakly-interactive mode.

Precise dating of late Neanderthal sites is still a delicate issue due to variability of radiocarbon dating and uncertainty of site stratigraphy. To achieve an approximation, we evaluated the data from [23] and references therein, [14] and [11] and compiled to our best knowledge a dataset of the MP/CHAT sites most likely younger than 42 ka. These are plotted in Fig 3c with the model simulated NEA population density for comparison. In contrast to the view that the southern part of the peninsula provided the last refuges for the NEAs, Fig 3c reveals that these are scattered in the (1) Cantabrian, (2) Portuguese, and (3) southern-tip, and (4) Mediterranean coastal areas. The model results are in general consistent with the location of the latest MP/CHAT sites in the North and Northeast.

The Cardina-Salto do Boi site (7.10°W, 40.97°N, Portugal) is highly interesting. According to [93], geomorphology, archaeostratigraphy, stone-tool evolution, and optically stimulated luminescence dating support the persistence of NEAs after 41 ka in this area. The Cardina site was not used for training the HEP model, but our model consistently predicted the presence of the NEA in the area in ExpNEA-H and ExpNEA-C (Fig 3a, 3b), acting as the missing link between northern coastal Iberia and the Atlantic coast of northern Portugal. The model suggests the Duero drainage system as a connecting corridor.

Vaesen et al. [87] argued that the NEA extinction might have resided in the smallness of their population alone, and can be explained by inbreeding, Allee effects and stochasticity. [89] conducted a survey on the hypotheses currently endorsed to explain the NEA extinction and reported that there is received wisdom that demography was the main cause of the NEA demise but no received wisdom about the role that environmental factors and competition with modern humans played in the process. Our findings support these statements in general. But why did the NEA population size become small in the first place? From the modeling perspective, decreases in $C$ and $r_B$ are necessary for the NEA population size to reach a critically low level for the mating network to break down, which may lead to the extinction of the entire population. In OW-CABM, mating network is accounted for in Eq 12. Based on the model simulations, a plausible explanation for the extinction of the NEAs is that the NEA population was already in decline for demographic reasons before the arrival of the AMHs, which resulted in small carrying capacity and low population growth rate, and the harsh climate conditions in the GS times further reduced the population size and the eventual breaking down of the mating network. HE5 appears to be a high-risks period for NEA extinction.

## Arrival and dispersal of Aurignacian in Iberia

We reconstructed the dispersal of the AMHs of the Aurignacian in Europe and predicted that the AMHs arrived in southern France and northern Iberia in the later expansion stage at 41.6 ± 0.67 ka [31], consistent with the Cantabrian sites dated to 43.3-40.5 ka [30]. Our reconstruction of AMH dispersal in Iberia using the OW-CABM is done based on this understanding, using the model parameters for the ExpAUR-C run (Table 2).

The ensemble- and time-averaged population density of the AMHs for selected times are shown in Fig 5. It is assumed that the AMHs arrived in southwest France at 42.5 ka. But for the initialization, a population center is specified at the Mediterranean coast of France at 43 ka (Fig 5a), and the model is allowed to spin-up for 500 years, during which humans first moved around the center and then started to disperse to southwest France and Franco-Cantabria, along the corridor to the north of the Pyrenees. By 42 ka, the Mediterranean coast and Atlantic coast of France, and Cantabria were populated (Fig 5b, 5c). With the onset of GS9/HE4, the settlement areas contracted and human expansion slowed (Fig 5d). Over the 3000 years of 41 to 38 ka, the basic features of the AMH population distribution remained similar. However, also in this period, the AMHs started to migrate along the Mediterranean coast from southwest France past the Pyrenees into the Ebro Valley. The model reveals a possible route of expansion of AMHs to Galicia not evident from the existing archaeological sites. Additional areas of possible expansion pop up later in the Duero area in northern Portugal and even south of the Iberian range. The model simulations suggest that around 41 ka, the AMHs in west Cantabria started the southward excursion, around 40 ka, appeared in northwest Portugal, and around 38 ka, reached south Sub-Meseta, crossed the Central System and entered the Madrid area. According to [93], the transition between the MP and the AUR is demarcated by two Optically Stimulated Luminescence (OSL) dates between 34.0 ± 2.0 ka and 38.4 ± 1.9 ka. While it can be argued whether the Aurignacian at Cardina belongs the AUR-1 or AUR-P2, the model prediction is highly interesting, as

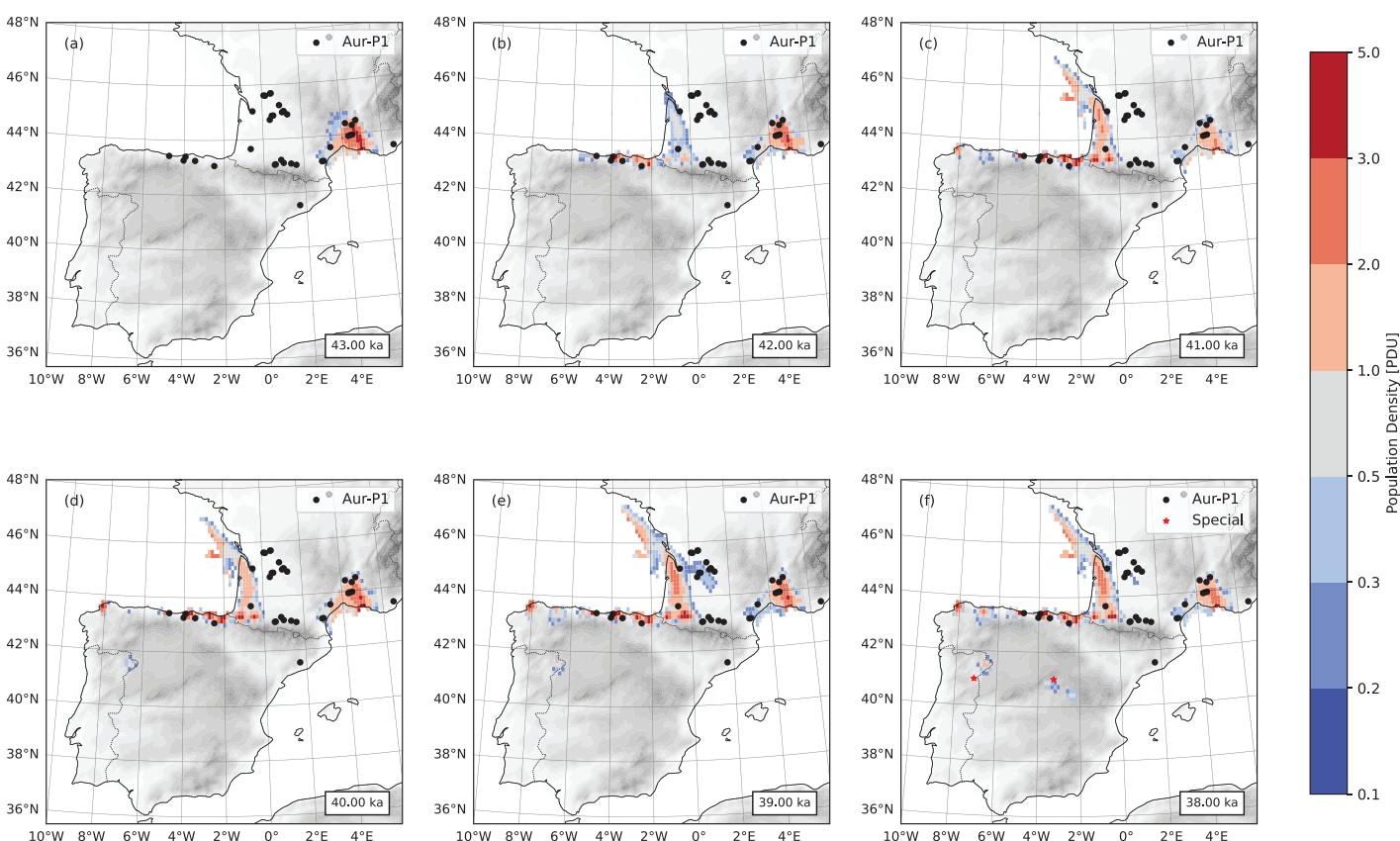

**Fig 5. Population density of the AMHs.** The population density (shaded) is ensemble-and-time averaged for the selected time slices, each of which represents a 100-year interval. Also plotted is the corresponding standard deviation showing the uncertainties of the averages. (a) Time slice 43 ka; (b), (c), (d), (e) and (f) as (a), but for 42, 41, 40, 39 and 38 ka, respectively. In (f), the sites of Cardina-Salto do Boi and Abrigo de la Malia are marked with red star.

Fig 5d, 5e and 5f show, in that it points to a new potential route of AUR expansion. Further more, in previous studies, no evidence was found of AMH occupation in central Iberia until 26 ka, rendering the region "nobody's land" during the entire Aurignacian. Recently, [94] found that Abrigo de la Malia provides irrefutable evidence of AMH settlements dating back to 36.20 to 31.76 ka. Here again we are at the boundary between AUR-P1 and AUR-P2 and the limits of the accuracy of radiocarbon dating. Our model predicts (without including the site in training the HEP model) that AMHs reached central Iberia earlier than the archaeological evidence we have so far, and it is consistent with [94] in general that we can expect at least temporal advances of the AMHs in the "nobody's land" and to northern and central Portugal via the "Duero Route".

**NEA and AMH admixture**

The OW-CABM is applied to simulating the NEA and AMH populations and the likely admixture in the weakly-interactive mode of the MUPT. The population of the NEA and AMH off springs is denoted as MIX to facilitate description. The model parameters are as specified in Table 2 for ExpNEA-C, ExpAUR-C and ExpMIX-C for the NEA, AMH and MIX populations, respectively. Admixing between the NEAs and AMHs is permitted with a probability of 1%.

The resulting MIX population can grow and disperse using parameters similar to those of the NEA and AMH populations, while its HEP is calculated as the arithmetic average of the NEA and AMH HEP values. Additional admixture possibilities beyond this initial crossing are not considered in this study. It should be clarified here that the admixture population is hypothetical under plausible model assumptions. So far no evidence has been found to suggest that admixing on the Iberian Peninsula took place or otherwise. Thus, the model-predicted admixture population pattern and size cannot be verified with any data and remain hypothetical and indicative.

Fig 6 shows the ensemble- and time-averaged (over period 42 - 38 ka) densities of the NEA, AMH and MIX populations. During the MUPT, the NEA and AMH populations were largely separated, with the NEAs living mostly in the coastal areas around the peninsula and in the submeseta area (Fig 6a) and the AMHs in the Atlantic coastal area of France and Cantabria and, with small probability, in the sub-mesata area (Fig 6b). Consequently, the regions of the admixture between the NEA and AMH populations were confined to Cantabria and, with small probability, a few isolated inland areas, as Fig 6c shows.

The ensemble-averaged size of the NEA, AMH and MIX populations is shown in Fig 7. Over the 7000-year period, based on the ExpNEA-C run, the NEAs maintained a population size between 500 and 1500 humans, which fluctuated with climate change. During GS times, the NEAs population size was kept at 600 humans on average. The GS9/HE4 climate shift reduced the NEA population by about 40%, from 1000 to 600. Upon arriving in southern France, the AMHs continued to expand rapidly, and correspondingly, the size of AMHs population increased strongly from about 400 to nearly 1700 humans in the period 42.5 - 41 ka. The GS9/HE4 climate shift almost halved the AMH population size. During the GS9/HE4 period, the AMH population size recovered slowly. The MIX population also experienced a growth period between 42.5 and 41 ka, from 0 to about 240 humans. With the drop of the NEA and AMH populations at the onset of GS9/HE4, the size of the MIX population halved. During the GS9/HE4 period, the size of the MIX population gradually recovered to about 200 humans, as a consequence of the admixture primarily in the Cantabria, accounting for 6% of the total. If the ExpNEA-L run is considered, then the MIX population size was below 30 humans, accounting for less than 2% of the total.

The dispersal characteristics of the NEA, AMH and MIX populations are as depicted in Fig 8 which shows the rate of population density change, estimated as $(\langle \rho(t + \Delta t) \rangle - \langle \rho(t) \rangle)/\Delta t$, with $\langle \rho(t) \rangle$ being the ensemble-averaged population density at time $t$ and $\Delta t$ a time interval of 500 years. In the first 1500 years or so, upon arriving at the Mediterranean coast of France, the AMHs continued to expand in a rapid wave to the Atlantic coast and Cantabria. In these regions, the population density substantially increased (Fig 8a). In the same period, the population at the Mediterranean coast of France decreased as the migration of humans out of the region outweighed the net population growth. With the onset of the

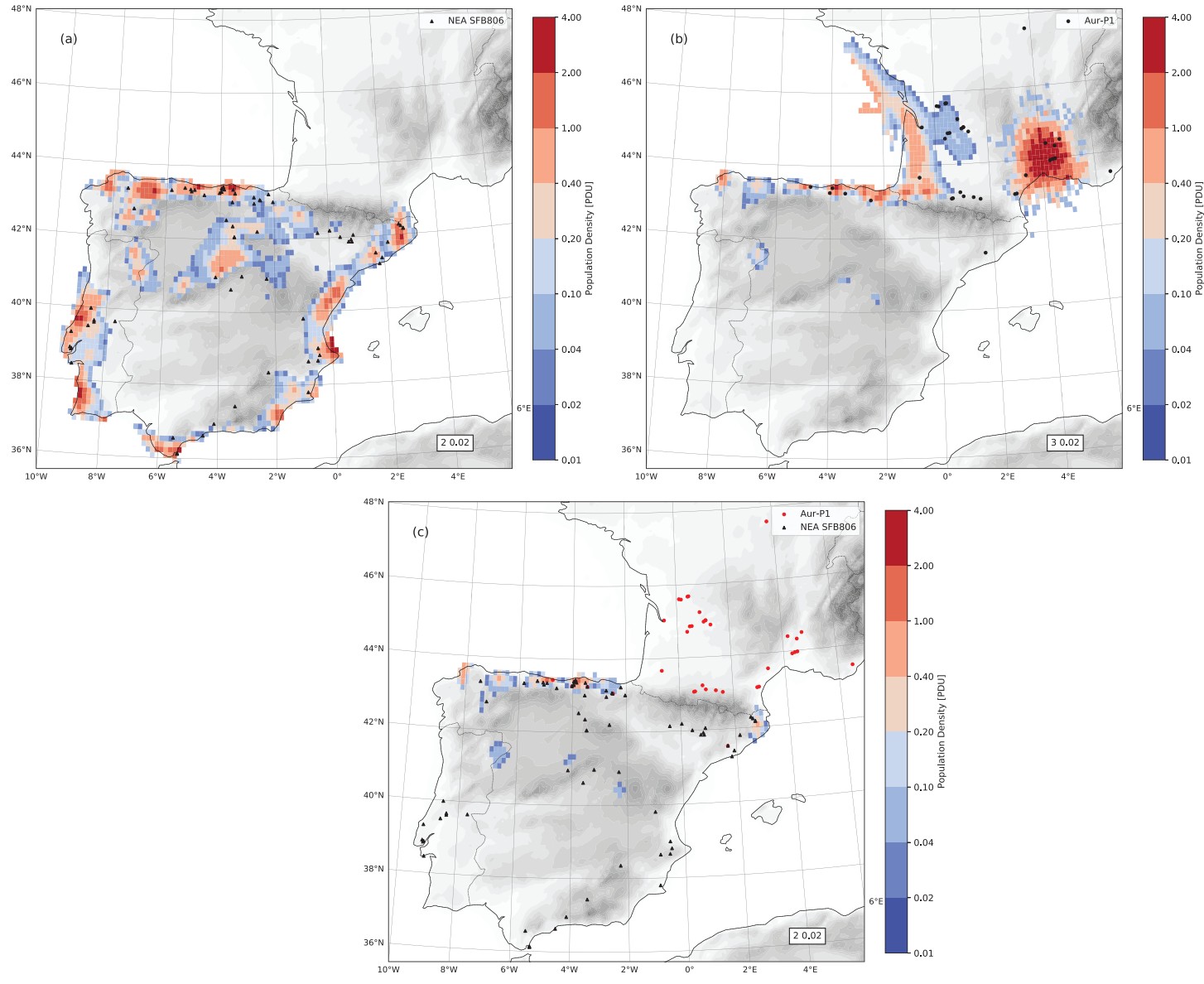

**Fig 6**. **Density of the NEA, AUR and MIX populations.** Plotted are the ensemble- and time-averages (over 42 - 38 ka) population density in PDU together with topography (grey shaded) and archaeological sites. (a) the NEA population and MP sites (triangles); (b) the AUR population and the AUR-P1 sites (black dots); and (c) for the MIX population and the MP sites (triangles) and AUR-P1 sites (red dots).

GS9/HE4 at around 41.15 ka, the AMH population density $\rho_{AHM}$ decreased sharply on the Atlantic coast of France and in Cantabria, apart from the hot spots. In contrast, $\rho_{AHM}$ increased on the Mediterranean coast of France, as humans flowed back to this region of high HEP (Fig 8b). This increase is important, as it sets the stage for the dispersal of the AMHs southwards around the Pyrenees into the Ebro Valley and further into the peninsula in the second phase of the Aurignacian. In the next 2000 years, the dispersal of the AMHs on the Iberian Peninsula is characterized by population fluctuations at the population centers (Mediterranean and Atlantic coast of France and Cantabria), continued exploration into the Ebro Valley and into the interior from the west end of the Cantabria (Fig 8b). In comparison, for the NEA and MIX populations there was no wave-like expansion. Instead, the population density mainly fluctuated in the populated regions

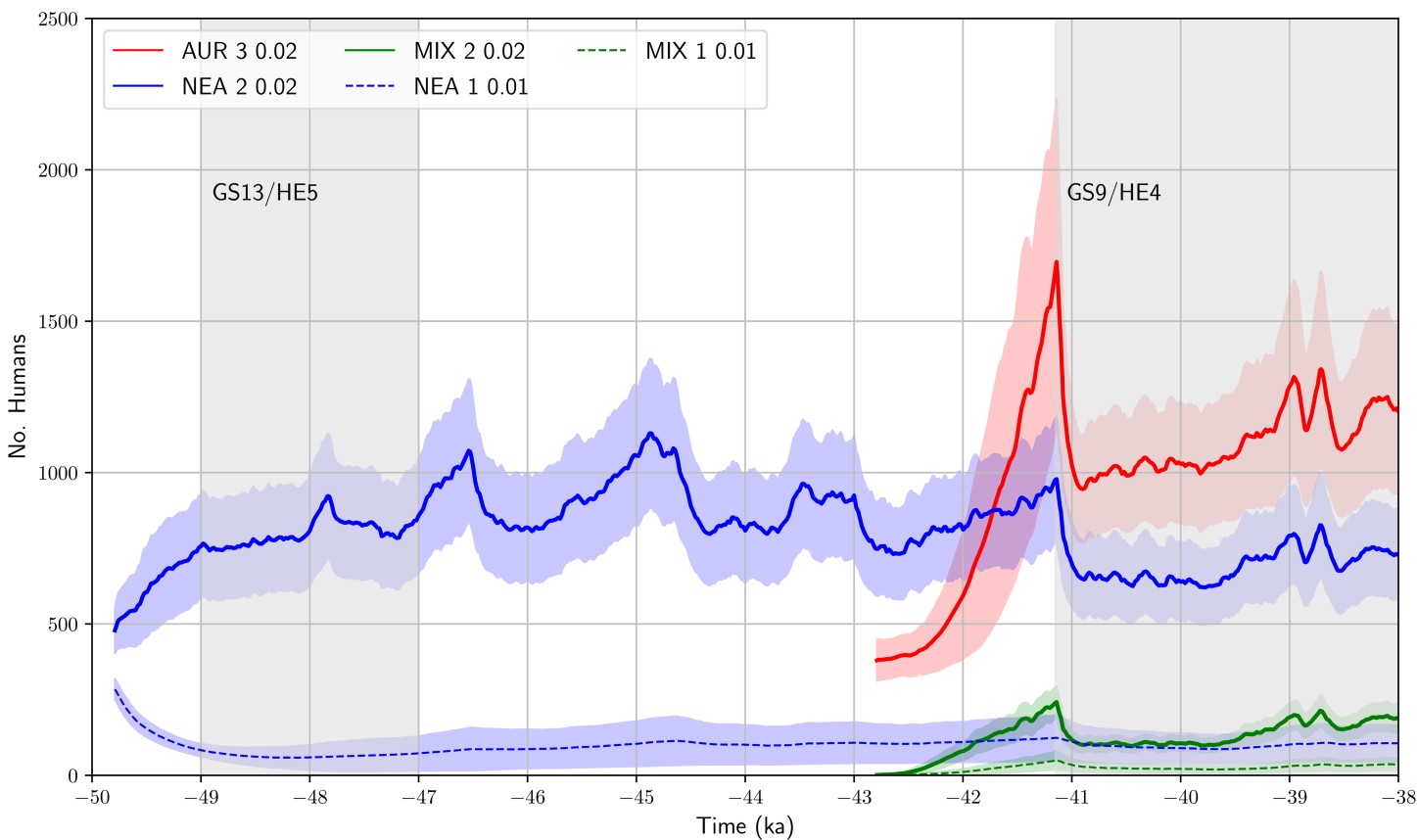

**Fig 7**. **Ensemble-mean AMH, NEA and MIX population size, in red, blue and green, respectively.** The thick line represents the ensemble-mean $P$ and shaded area $P \pm \sigma_P$ with $\sigma_P$ being the ensemble standard deviation. Also plotted are the NEA and MIX population, in blue and green dashed lines, respectively, corresponding to the ExpNEA-L run.

(Fig 8d). With the GS9/HE4 climate shift, the NEA population density $\rho_{NEA}$ significantly decreased, with humans retreating from more inland areas to the coastal refuges, where (e.g., west most Cantabria) $\rho_{NEA}$ increased (Fig 8e, 8f). The population density changes of the MIX population took place mainly in the regions of AMH and NEA admixture (Fig 8g). With the GS9/HE4 climate shift, as a result of the NEA and AMH population decreases in Cantabria, the chances of NEA and AMH interbreeding reduced, causing a significant decrease in the MIX population density (Fig 8h, 8i).

## Lapa do Picareiro

The findings from Lapa do Picareiro in Portugal [13] suggest that AMHs reached here already in AUR-P1, although the assignment of the site to the AUR-P1 has been disputed [14]. At Lapa do Picareiro, the small stone tool inventory of less than 50 pieces can be assigned to an Aurignacian, but the early radiocarbon dates all come from the area below or from the lower edge of the find layer. Therefore, it cannot be decided with certainty that they actually date the find context. The findings of Lapa do Picareiro, if confirmed, would change the common view on the AMH settlement history in Iberia, at least in its western part.

On the pan-European scale with 50 km spatial resolution, the model simulations of the Aurignacian dispersal suggested that the AMHs of the Aurignacian reached the west coast of Portugal in AUR-P2 [31]. However, the high-resolution (12.5 km) HEP analysis [11] showed that there existed a potential corridor weakly linking the Lapa do Picareiro area

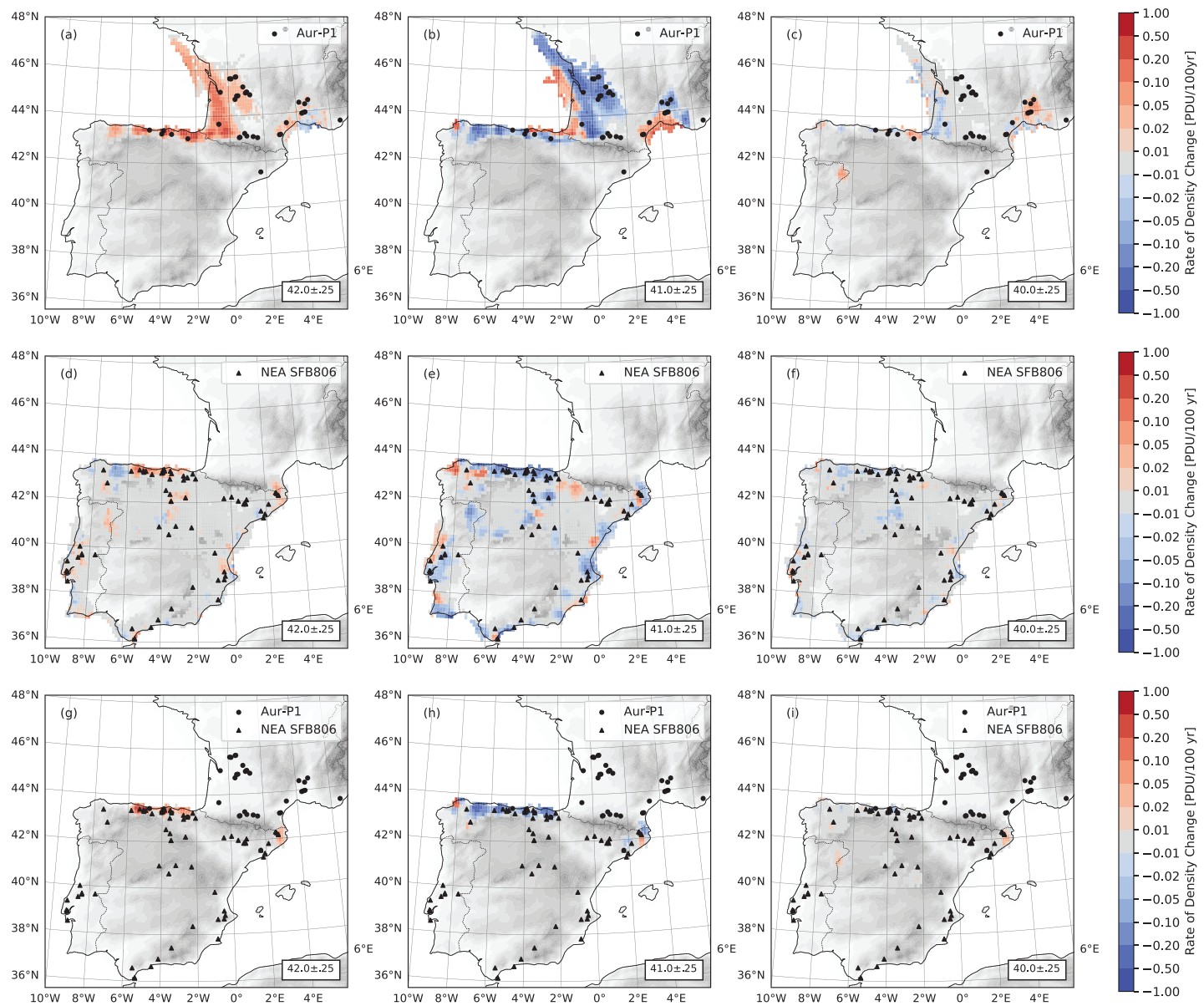

**Fig 8**. **Rate of population density change in PDU/100 yr.** (a), (b) and (c) for AMH, (d), (e) and (f) for NEA population, and (g), (h) and (i) for MIX population at 42, 41 and 40 ka, respectively.

to Franco-Cantabria via west Spain. Thus, the AMH settlement at Lapa do Picareiro was not impossible, although an event of small probability. Our new simulation of the AMH dispersal with the Lapa do Picareiro site considered in the HEP calculation is as shown in Fig 5.

For the period 43 - 41 ka, the simulated AMH population distributions, as seen in Figs 5 and 9, are identical, but for the period 40 - 38 ka different. With the Lapa do Picareiro site included, the model simulated the intrusion of AMHs to the western coast of Portugal at around 40 ka. They AMHs could have reached here from the Franco-Cantabria region via

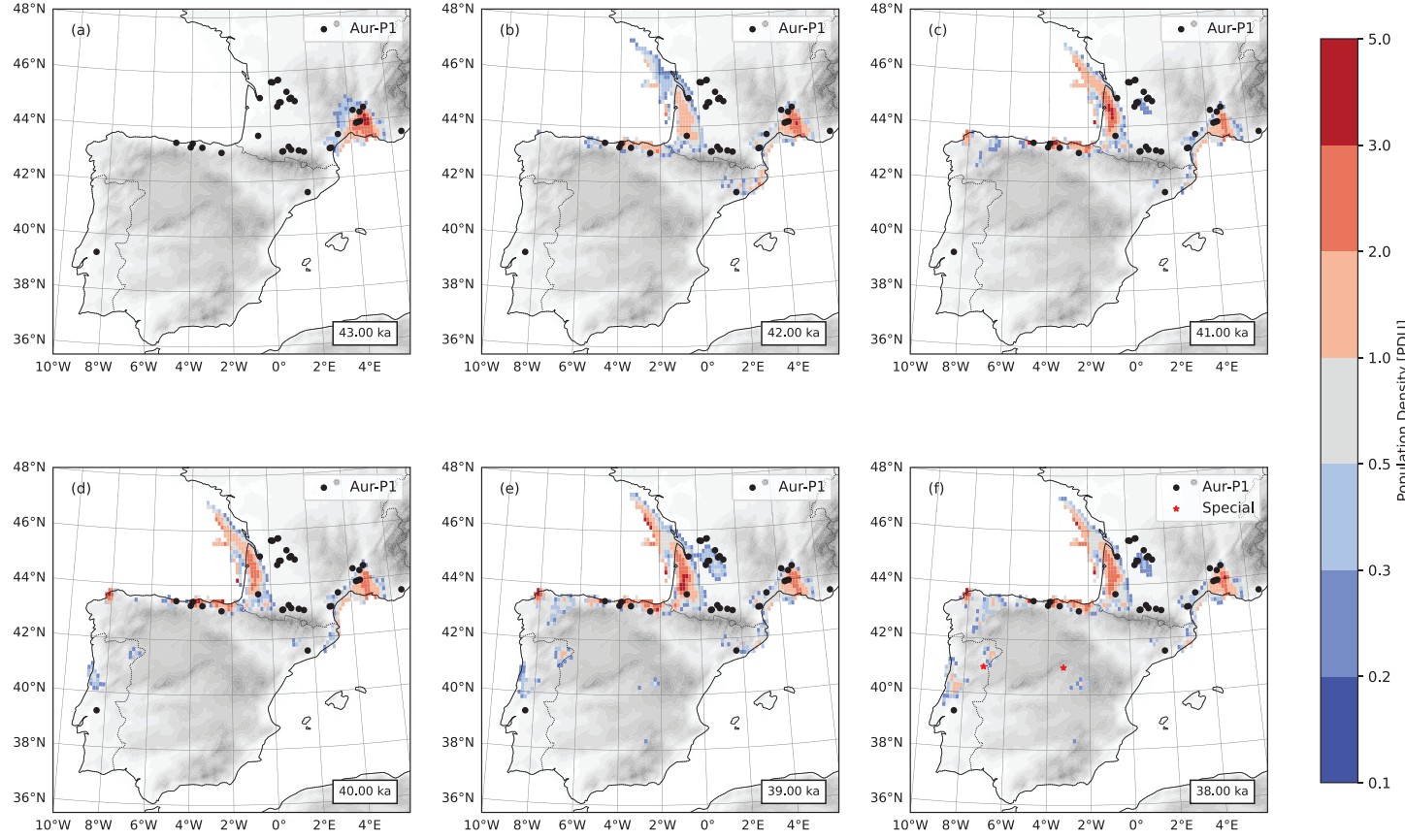

**Fig 9. As Fig 5, but HEP is estimated with the Lapa do Picareiro site included.**

the western part of Spain. The AMHs probably existed in separation with the main population centers in France and north Spain and their occupation area only slightly enlarged over the next 2000 years (Fig 9f).

## Conclusions

By modeling the population networks, dispersal dynamics, and admixture processes of Neanderthals (NEAs) and anatomically modern humans (AMHs) under a weakly interactive scenario, our simulations provide a comprehensive overview of the Middle to Upper Paleolithic Transition (MUPT) in Iberia, as depicted in Fig 10.

We found that while millennial-scale climatic events such as Heinrich Event 5 (HE5) and Greenland Stadial 9/Heinrich Event 4 (GS9/HE4) substantially reduced NEA population sizes, these events alone were insufficient to cause extinction unless the NEA populations were already demographically fragile. Refugial zones—including Cantabria, Portugal, the southern tip of Iberia, and the Mediterranean coastline—could have buffered NEAs against complete population collapse. Assuming stable cultural carrying capacity and population growth rates, NEAs that survived HE5 might also have persisted through GS9/HE4. These findings support the prevailing interpretation that the NEA extinction resulted from a confluence of demographic instability and climatic stressors, consistent with the synthesis presented in [89].

The spatial distribution of Middle Paleolithic sites was well reproduced in the ExpNEA-H run using a cultural carrying capacity of $C = 3$ PDU and a birth rate $r_B = 0.05$ yr$^{-1}$. This suggests that NEAs initially maintained a robust capacity for

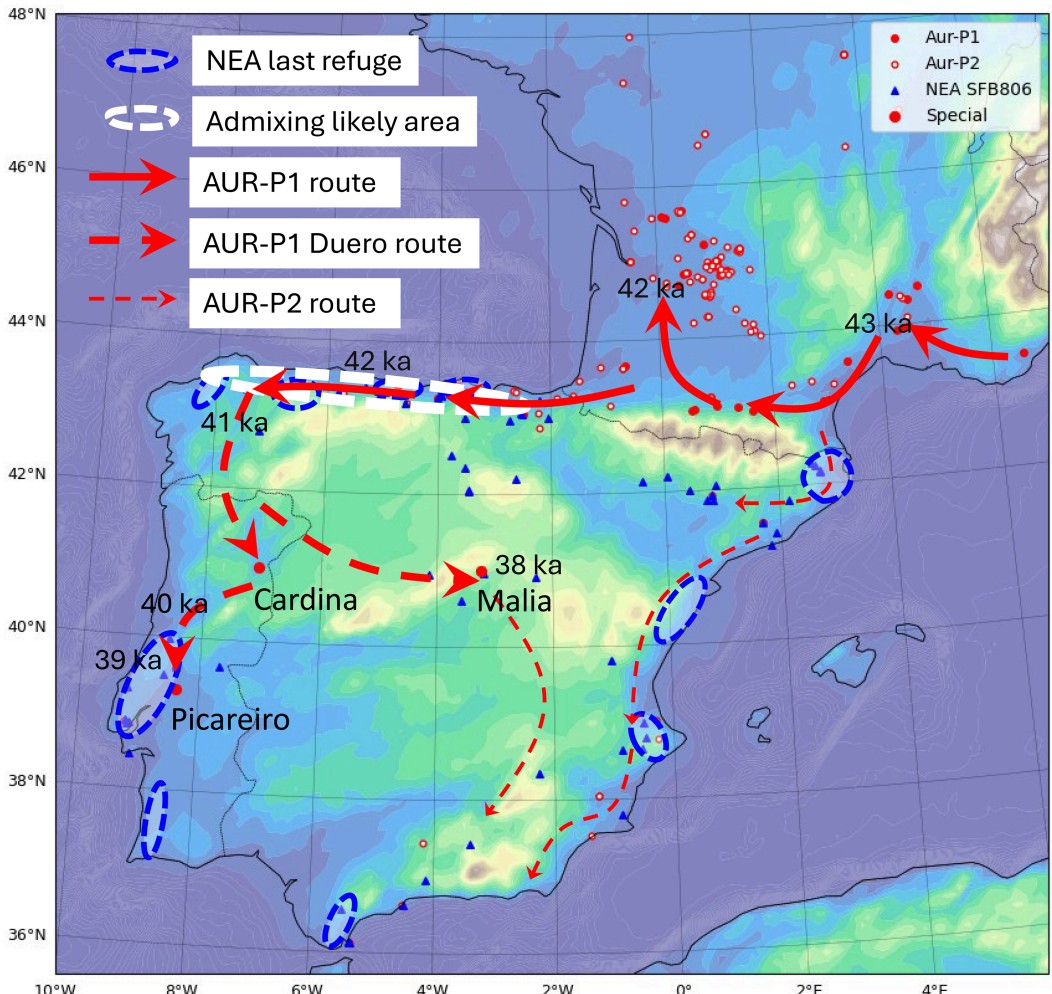

**Fig 10**. **Displayed are the MP (blue triangle), AUR-P1 (solid red circle), and AUR-P2 sites (open red circle).** Three special Aurignacian sites — Cardina-Salto do Boi, Abrigo de la Malia, and Lapa do Picareiro — are also indicated. The last Neanderthal (NEA) refugia are primarily located in coastal areas, delineated by a dashed blue circle. The initial expansion routes of anatomically modern humans (AMHs) associated with AUR-P1 are shown with thick solid red arrows. The subsequent Duero route, extending from Cantabria into inland Iberia, is represented by thick dashed red arrows. The expansion pathways of AMHs linked to AUR-P2 are marked by thin dashed red arrows. A potential zone of admixture between NEAs and AMHs is identified along the northern Iberian coast, enclosed by a white dashed circle.

population growth. However, their eventual extinction likely stemmed from a progressive and systematic loss of reproductive viability. This demographic decline, exacerbated by climatic stress, drove NEA populations to critically low levels at which mating networks disintegrated, ultimately leading to extinction. Based on our simulations, attributing NEA extinction solely to AMH expansion is unwarranted. Notably, the ExpNEA-L results indicate that HE5 was a particularly high-risk period for NEAs.

During a wave of AMH expansion, modern humans reached northern Spain, an area already settled by NEAs. In the weakly interactive scenario, this region represents the most plausible zone of NEA–AMH admixture. The size of the admixed population fluctuated with climate-induced variations in both NEA and AMH population sizes, ranging from 10 to 100 individuals and accounting for approximately 2–6% of the total population during the MUPT. While these model-derived estimates are currently unverified by empirical data, they offer testable hypotheses for future studies.

Archaeological evidence suggests that human migrations often followed river systems; our model supports this behavior by demonstrating preferential migration along corridors of high Human Existence Potential (HEP) values, which river valleys characteristically offer. Our simulations revealed a previously unnoticed inland dispersal corridor for AMHs. Although the GS9/HE4 climatic event interrupted their rapid and large-scale expansion, AMHs continued to move from western Cantabria through the Minho River Valley into the Duero drainage basin, even crossing the Central System and reaching further south—likely as temporary excursions. A westward branch of this movement opened a migration route into Portugal, enabling stable settlement and accounting for the presence of AMHs at the Lapa do Picareiro site (AUR-P1). An eastward path led into the so-called "nobody's land" of central Spain, near Abrigo de la Malia [94]. This newly identified Duero corridor explains the geographic context of the contiguous Picareiro site and supports its classification within the AUR-P1 complex.

During the AUR-P2 phase, AMH dispersal primarily followed the Mediterranean coastal route, with a likely secondary inland pathway extending southward from central Spain. It is even plausible to hypothesize that, after a temporal and spatial separation of nearly 5000 years, AMHs from these two fronts converged in southern Iberia.

Understanding human origin and dispersal has been challenged by fundamental limitations in traditional data sources. Archaeological and paleogenomic data are fragmented in space and time, provide limited demographic context, and are often disconnected from cultural and climatic/environmental records. These constraints restrict their full potential and leave numerous contradictions unresolved. To overcome these limitations, the OW-CABM framework presented here systematically integrates climatic/environmental, archaeological, and potentially cultural data through population dynamics modeling. By employing HEP as a unified and generalizable metric that drives demographic growth and spatial dispersal, our approach reconstructs hominin population expansion, interaction, and distribution across landscapes at regional and continental scales. This work offers a powerful alternative for investigating human origin and dispersal, that complements and contextualizes traditional archaeological and genomic approaches.

## Acknowledgments

We thank Guohao Liang for supporting the preparation of the model input data and Philipp Schlüter for preparing some of the python scripts used in data analysis.

## Supporting information

**S1 Appendix. Supplementary discussion and figures.** Includes further details for the Human Existence Potential, climate and archaeology timelines and model sensitivity tests.
(PDF)

## Author contributions

**Conceptualization:** Yaping Shao, Gerd-Christian Weniger.

**Data curation:** Konstantin Klein, Christian Wegener, Gerd-Christian Weniger.

**Formal analysis:** Yaping Shao.

**Funding acquisition:** Yaping Shao.

**Investigation:** Yaping Shao, Konstantin Klein, Christian Wegener, Gerd-Christian Weniger.

**Methodology:** Yaping Shao, Konstantin Klein, Christian Wegener, Gerd-Christian Weniger.

**Project administration:** Yaping Shao.

**Resources:** Yaping Shao.

**Software:** Yaping Shao, Konstantin Klein, Christian Wegener.

**Supervision:** Yaping Shao, Gerd-Christian Weniger.

**Validation:** Yaping Shao, Christian Wegener, Gerd-Christian Weniger.

**Visualization:** Yaping Shao, Konstantin Klein, Christian Wegener.

**Writing – original draft:** Yaping Shao.

**Writing – review & editing:** Yaping Shao, Konstantin Klein, Christian Wegener, Gerd-Christian Weniger.

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
