## [Decision Letter · Decision Letter 0]

20 Aug 2025

PONE-D-25-33066Pathways at the Iberian Crossroads: Dynamic

Modeling of the Middle–Upper Paleolithic TransitionPLOS ONE

Dear Dr. Shao,

Thank you for submitting your manuscript to PLOS ONE. After careful consideration, we feel that it has merit but does not fully meet PLOS ONE’s publication criteria as it currently stands. We received only one reviewer's report which however I consider fair and collaborative in contributing to strengthen your work and above all giving it the possibility to be reproducible. Furthermore, I fully agree with all points raised, particularly about the inclusion of Chatelperronian sites.Therefore, we invite you to submit a revised version of the manuscript that addresses the points raised during the review process.

We look forward to receiving your revised manuscript.

Kind regards,

Marco Peresani

Academic Editor

PLOS ONE

Journal Requirements:

2. In your manuscript, please provide additional information regarding the specimens used in your study. Ensure that you have reported human remain specimen numbers and complete repository information, including museum name and geographic location.

For more information on PLOS ONE's requirements for paleontology and archeology research, see https://journals.plos.org/plosone/s/submission-guidelines#loc-paleontology-and-archaeology-research.

“German Research Council; NRW Ministry of Science and Culture”

6. Thank you for stating the following in your Competing Interests section: 

“No competing interests”

7. Please provide a complete Data Availability Statement in the submission form, ensuring you include all necessary access information or a reason for why you are unable to make your data freely accessible. If your research concerns only data provided within your submission, please write "All data are in the manuscript and/or supporting information files" as your Data Availability Statement.

8. PLOS requires an ORCID iD for the corresponding author in Editorial Manager on papers submitted after December 6th, 2016. Please ensure that you have an ORCID iD and that it is validated in Editorial Manager. To do this, go to ‘Update my Information’ (in the upper left-hand corner of the main menu), and click on the Fetch/Validate link next to the ORCID field. This will take you to the ORCID site and allow you to create a new iD or authenticate a pre-existing iD in Editorial Manager.

9. We note that Figures 3,5,6,8,9,10 and supporting information figures 11,12,14,15,16 and 17 in your submission contain map/satellite images which may be copyrighted. All PLOS content is published under the Creative Commons Attribution License (CC BY 4.0), which means that the manuscript, images, and Supporting Information files will be freely available online, and any third party is permitted to access, download, copy, distribute, and use these materials in any way, even commercially, with proper attribution. For these reasons, we cannot publish previously copyrighted maps or satellite images created using proprietary data, such as Google software (Google Maps, Street View, and Earth). For more information, see our copyright guidelines: http://journals.plos.org/plosone/s/licenses-and-copyright.

 1. You may seek permission from the original copyright holder of Figures 3,5,6,8,9 and 10 to publish the content specifically under the CC BY 4.0 license. 

10. Please upload a copy of Figure 13, to which you refer in your text on page 2. If the figure is no longer to be included as part of the submission please remove all reference to it within the text.

11. Please include a copy of Table 4 and 5 which you refer to in your text on page 11.

12. Please include captions for your Supporting Information files at the end of your manuscript, and update any in-text citations to match accordingly. Please see our Supporting Information guidelines for more information: http://journals.plos.org/plosone/s/supporting-information.

Reviewers' comments:

Reviewer's Responses to Questions

**Comments to the Author**

1. Is the manuscript technically sound, and do the data support the conclusions?

Reviewer #1: Yes

2. Has the statistical analysis been performed appropriately and rigorously?

Reviewer #1: Yes

3. Have the authors made all data underlying the findings in their manuscript fully available?

Reviewer #1: No

4. Is the manuscript presented in an intelligible fashion and written in standard English?

Reviewer #1: Yes

5. Review Comments to the Author

Reviewer #1: This paper presents a new constrained agent-based model incorporating biological, cultural and dispersal processes to simulation Neanderthal and modern human population distributions in Iberia during the Middle to Upper Paleolithic transition (MUPT). Building on previously published Human Existence Potential (HEP) models by Klein et al (2023), the authors explore population movement out of centers of occupation and across the Iberian Peninsula. Key contributions include the identification of potential dispersal routes into and within Iberia, a hypothesized location for Neanderthal-modern human interbreeding, and support for a coupled demographic-climate model for Neanderthal extinction.

While the constrained ABM is generally well-explained and justified, the absence of code access and the lack of sensitivity analyses addressing key assumptions limit the reproducibility and robustness of the results. For these reasons, I recommend major revisions.

Major comments:

The paper assumes that the HEP model provides a reasonable estimate of suitable places for Neanderthal and modern human occupation. While the HEP is robust based on training methods, its reliability might be compromised by limiting the training data to sites from the Iberian Peninsula (Barr and Wood, 2024). Have the authors tested whether excluding sites from the broader European continent significantly alters the results? The authors’ own prior paper (Shao et al 2021) shows different HEP for the Iberian Peninsula interstadial periods when using a pan-European site list. The influence of site inclusion is also evident in this paper’s results – adding the Picareiro changed model results and interpretations. The authors should either: 1) justify restricting the site list to the Iberian Peninsula or 2) demonstrate that using a broader dataset would not significantly affect model results.

The inclusion of Châtelperronian sites in the Middle Paleolithic sample representing Neanderthal presence is problematic given unresolved debates about authorship (Gravina et al., 2018; Djakovic et al., 2024). The discussion of the Châtelperronian in the introduction is too brief consider the debates surrounding this industry. The authors should make clear here how they are treating the Châtelperronian sites because the way it is currently written seems to suggest that these sites would be the location of “mixed” populations. Additionally, same as the above point, the authors should demonstrate that removing the Châtelperronian sites would not change model results.

Neanderthal simulations begin from four geographic points, but no rationale is provided for these locations. Presumably, these correspond to archaeological sites, but this should be explicitly stated. Further, the authors should test how variation in number, location, and distribution of seed sites affects results.

The use of N0 = 1000 for Neanderthals is not explained or tested. Although the authors dismiss 2000-3000 individuals as too large a population in their results section, this contradicts effective population size estimates (e.g. Lalueza-Fox et al., 2005; Bocquet-Appel and Degioanni, 2013; Prüfer et al., 2014), where 1000 individuals is on the lower end. The authors should justify this parameter using archaeological and/or genetic data. The authors should also test the sensitivity of the model to different N0 values. The same applies to modern human initial population size, which is left unexplained and untested.

Minor comments:

• The introduction references network formation but this is only considered in population growth (equation 8). If true network dynamics are not modeled, either remove this framing or expand on how networks function in the simulation.

• Description of the HEP variants is fragmented. Their mathematical derivation appears under population growth but would be clearer if presented with the initial definitions. Additionally, I suggest reorganizing Sections 2.4 and 2.5 (input data for HEP calculations) to precede the growth and mobility models, so that the logic flows from environmental modeling to demographic modeling to mobility modeling to model parameter values.

• Modern human seeding is taken from the results of a previous paper, but that study used a different dispersal model. The justification for using this location should be clarified and supported.

• The interbreeding dynamics should be clarified: do admixed individuals reproduce independently, or only through additional interbreeding events?

• The ending feels abrupt. Consider adding a statement about why this kind of modeling is relevant or future directions for empirical validation.

• Some main text references point to Supplement figures/tables. These should be clearly annotated.

• There are a few equations where variables are not defined in the text: for example, no definition of rB for equation 7, no definition of m for equation 15.

• In Supplementary Material:

In Section 8.1, should the results for T be plotted somewhere?

In Section 8.2, should the cb be alpha?

In site table, it would be good to specify what sites have MP vs CHÂT.

References:

Barr, W.A., Wood, B., 2024. Spatial sampling bias influences our understanding of early hominin evolution in eastern Africa. Nature Ecology & Evolution.

Bocquet-Appel, J.-P., Degioanni, A., 2013. Neanderthal Demographic Estimates. Current Anthropology. 54, S202–S213.

Djakovic, I., Roussel, M., Soressi, M., 2024. Stone Tools in Shifting Sands: Past, Present, and Future Perspectives on the Châtelperronian Stone Tool Industry. Journal of Paleolithic Archaeology. 7, 29.

Gravina, B., Bachellerie, F., Caux, S., Discamps, E., Faivre, J.-P., Galland, A., Michel, A., Teyssandier, N., Bordes, J.-G., 2018. No Reliable Evidence for a Neanderthal-Châtelperronian Association at La Roche-à-Pierrot, Saint-Césaire. Scientific Reports. 8, 15134.

Lalueza-Fox, C., Sampietro, M.L., Caramelli, D., Puder, Y., Lari, M., Calafell, F., Martínez-Maza, C., Bastir, M., Fortea, J., Rasilla, M. de la, Bertranpetit, J., Rosas, A., 2005. Neandertal Evolutionary Genetics: Mitochondrial DNA Data from the Iberian Peninsula. Molecular Biology and Evolution. 22, 1077–1081.

Prüfer, K., Racimo, F., Patterson, N., Jay, F., Sankararaman, S., Sawyer, S., Heinze, A., Renaud, G., Sudmant, P.H., de Filippo, C., Li, H., Mallick, S., Dannemann, M., Fu, Q., Kircher, M., Kuhlwilm, M., Lachmann, M., Meyer, M., Ongyerth, M., Siebauer, M., Theunert, C., Tandon, A., Moorjani, P., Pickrell, J., Mullikin, J.C., Vohr, S.H., Green, R.E., Hellmann, I., Johnson, P.L.F., Blanche, H., Cann, H., Kitzman, J.O., Shendure, J., Eichler, E.E., Lein, E.S., Bakken, T.E., Golovanova, L.V., Doronichev, V.B., Shunkov, M.V., Derevianko, A.P., Viola, B., Slatkin, M., Reich, D., Kelso, J., Pääbo, S., 2014. The complete genome sequence of a Neanderthal from the Altai Mountains. Nature. 505, 43–49.

6. PLOS authors have the option to publish the peer review history of their article (what does this mean?). If published, this will include your full peer review and any attached files.

Reviewer #1: No

---

## [Author Response · Author response to Decision Letter 1]

7 Nov 2025

Review Comments to the Author

Reviewer #1: This paper presents a new constrained agent-based model incorporating biological, cultural and dispersal processes to simulation Neanderthal and modern human population distributions in Iberia during the Middle to Upper Paleolithic transition (MUPT). Building on previously published Human Existence Potential (HEP) models by Klein et al (2023), the authors explore population movement out of centers of occupation and across the Iberian Peninsula. Key contributions include the identification of potential dispersal routes into and within Iberia, a hypothesized location for Neanderthal-modern human interbreeding, and support for a coupled demographic-climate model for Neanderthal extinction.

While the constrained ABM is generally well-explained and justified, the absence of code access and the lack of sensitivity analyses addressing key assumptions limit the reproducibility and robustness of the results. For these reasons, I recommend major revisions.

We thank Referee 1 for their careful review of our study and for the in-depth and constructive comments.

As stated in our "Data and Materials Availability" section, all data and model code will be made available in a suitable non-profit online repository that issues a DOI, with no restrictions on sharing or re-use. We have now uploaded the data, including archaeological site data, human existence potentials for AMH, NEA, and MIX populations, and the model code, for public use. These materials can be accessed via the data repository doi:10.6084/m9.figshare.30529043.

We have, in fact, conducted several hundred sensitivity experiments to address the key assumptions of our study. The sensitivity tests on the Human Existence Potential (HEP) analysis were presented in our earlier publications, and the Constrained Agent-Based Model (CABM) has been extensively tested. A sample of eight sensitivity experiments has been presented and discussed in the "Sensitivity to Model Parameters" section of Supplementary Materials. Numerous other experiments are detailed in Klein's 2022 PhD thesis, "Simulating Paleolithic Human Dispersal using Human Existence Potential and Constrained Random Walk Model" (University of Cologne).

In the revised manuscript, we have added a clarification to better highlight the sensitivity tests we have performed. Furthermore, as outlined below, we have included additional sensitivity tests in the revised version that examine the effects of adding Châtelperronian sites and varying the initial population distribution.

Major comments:

The paper assumes that the HEP model provides a reasonable estimate of suitable places for Neanderthal and modern human occupation. While HEP is robust based on training methods, its reliability might be compromised by limiting the training data to sites from the Iberian Peninsula (Barr and Wood, 2024). Have the authors tested whether excluding sites from the broader European continent significantly alters the results? The authors’ own prior paper (Shao et al 2021) shows different HEP for the Iberian Peninsula interstadial periods when using a pan-European site list. The influence of site inclusion is also evident in this paper’s results – adding the Picareiro changed model results and interpretations. The authors should either: 1) justify restricting the site list to the Iberian Peninsula or 2) demonstrate that using a broader dataset would not significantly affect model results.

We appreciate the referee raising the issue of HEP sensitivity with adding or removing sites. The robustness of the HEP model has been evaluated through extensive tests (Klein et al. 2021, Klein et al. 2023, Shao et al. 2021, Shao et al. 2024). In our recent unpublished work for southern Africa, almost exhaustive tests have been done. All these tests consistently confirm the model’s stability under reasonable perturbations. In the ensemble approach of training the model 1,000 times (each on an 80% random sample of archaeological data, equivalent to a 20% omission per run), the reliability of HEP estimates is quantified. While the ensemble mean gives the final HEP estimates we use for dynamic modelling, the ensemble standard deviation serves as a measure of uncertainty. In a pan-European application modeling AUR-P1 AMHs, Shao et al. (2024) [their Fig. A3c] showed that the standard deviation is generally low, indicating high consistency across most of the domain. However, elevated uncertainty was observed in certain regions, including central Iberia and the northern periphery of the human settlement range. Interpretation of HEP in these areas should be treated with caution.

Although the HEP model is robust, generating optimal estimates for population dynamics is still hampered by the spatial heterogeneity of archaeological data, alongside uncertainties in paleoclimate reconstructions. This aligns with Barr and Wood (2024, doi.10.1038/s41559-024-02522-5)’s caution that spatially biased sampling (e.g., from the limited Eastern African Rift) distorts interpretations, as their data cannot represent Africa's full environmental diversity. In Europe for AMHs, where site coverage is more representative, HEP patterns are largely resilient to the addition or removal of a limited fraction of sites.

We acknowledge that exceptions are sites offering novel information, such as Portugal's Lapa do Picareiro. Isolated hundreds of kilometers from the main AUR-P1 cluster, its inclusion in HEP training can somewhat shift HEP estimates for Iberia. While weak and spatially limited, these shifts have demographic implications, especially given the site's contested status, a topic we have discussed using a dedicated section “Lapa do Picareiro”.

Our Iberian HEP estimates for AMHs, derived from a localized archaeological dataset and high-resolution (12.5 km) climate data, exhibit more detailed spatial variations compared to the pan-European results of Shao et al. (2021, 2024), which were derived using a broad spatiotemporal dataset and low-resolution (50 km) climate data. While the pan-European approach captures large-scale HEP patterns, its reliance on data spanning several millennia limits its suitability for the high-resolution demographic modeling intended in this study. For instance, the low-resolution data obscured subtle regional population connections within Iberia, such as the weak demographic link between the northern coast and inland regions. Consequently, by adopting a regional high-resolution approach, we acknowledge that methodological differences may lead to quantitative divergences in results.

We have, as the Referee suggests, added a new section on HEP modelling and added the above argument to the text.

The inclusion of Châtelperronian sites in the Middle Paleolithic sample representing Neanderthal presence is problematic given unresolved debates about authorship (Gravina et al., 2018; Djakovic et al., 2024). The discussion of the Châtelperronian in the introduction is too brief consider the debates surrounding this industry. The authors should make clear here how they are treating the Châtelperronian sites because the way it is currently written seems to suggest that these sites would be the location of “mixed” populations. Additionally, same as the above point, the authors should demonstrate that removing the Châtelperronian sites would not change model results.

The Châtelperronian (Chât) topic is indeed somewhat tricky because its distribution area overlaps with that of the AUR 1, which can give the impression that the Chât and mixed populations are identical. Overall, the assessment of the Chât is highly controversial and has not yet been settled within the archaeological community. This is due to the small number of sites in Northern Spain and the limited chronological resolution, and like the AUR 1, it comes from France to the Iberian Peninsula. Therefore, providing an answer is difficult. As suggested by the Referee, we have computed HEP for the NEA population with and without the Chât sites. The results are presented in a new section in the Supplement. As expected, compared with the HEP results computed with the Chât sites, no significant difference can be seen in the HEP patterns, only minor differences in the HEP magnitude can be detected. These minor differences have little impact on the model conclusions. This test also and again demonstrates that omitting a small set of sites does not lead to significant changes in the HEP results.

Neanderthal simulations begin from four geographic points, but no rationale is provided for these locations. Presumably, these correspond to archaeological sites, but this should be explicitly stated. Further, the authors should test how variation in number, location, and distribution of seed sites affects results.

The use of N0 = 1000 for Neanderthals is not explained or tested. Although the authors dismiss 2000-3000 individuals as too large a population in their results section, this contradicts effective population size estimates (e.g. Lalueza-Fox et al., 2005; Bocquet-Appel and Degioanni, 2013; Prüfer et al., 2014), where 1000 individuals is on the lower end. The authors should justify this parameter using archaeological and/or genetic data. The authors should also test the sensitivity of the model to different N0 values. The same applies to modern human initial population size, which is left unexplained and untested.

The in-depth comment of the Referee is greatly appreciated. Some uncertainties exist with the model initialization. These are dealt with to allow the model to have a spin up time, here set to 500 years. Our unpublished tests show that the model outcomes are not too sensitive to the initialization. To fully address these issues, we conducted a set of sensitivity tests and included the results in a dedicated supplementary section.

As we have stated, for model initialization, N0 humans are Gaussian distributed in population centers with prespecified mean positions and standard deviations. For NEAs, 1000 humans are Gaussian distributed with a standard deviation σ of 1° at four locations at (3.93°W, 42.3°N), (0.72°W, 40.03°N), (5.54°W, 36.66°N) and (8.6°W, 39.2°N). To answer the questions of the Referee, it is sufficient to vary N0 and σ. The meaning of varying N0 is self-explained. By varying σ the spread of humans is changed. A large σ is equivalent to a more random population distribution. The sensitivity experiments are listed in the table below. In Exp 3, σ is enlarged by factor of 3, but with this, some of the initial seed humans will land in ocean and excluded from simulation, therefore a larger N0 ( = 3000 ) is also set for this experiment. In Exp 4 and Exp 5, we randomly selected 4 other population centers for testing. The tests show that the initial condition significantly influences the model simulation in the initial 500 to 1000 years. Beyond this, all test runs give very similar results with the variance between the tests much smaller than that of the ensemble runs for each test (see figure below). We added a new section to the supplementary materials to summarize the test results.

Tests

(File Name) N0 L1 L2 L3 L4 σ

Control (N01000) 1000 3.93°W, 42.3°N 0.72°W, 40.03°N 5.54°W, 36.66°N 8.6°W, 39.2°N 1°

Exp 1

(N0500) 500 - - - - -

Exp 2

(N02000) 2000 - - - - -

Exp 3

(N03000) 3000 - - - - 3

Exp4

(N01000a) 1000 8.0°W, 41.0°N 2.0°W, 42.0°N 2.0°W, 39.0°N 6.0°W, 38.0°N 3

Exp5

(N01000b) 1000 - - - - 1

We reviewed estimates of NEA population size from Lalueza-Fox et al. (2005), Bocquet-Appel and Degioanni (2013), and Prüfer et al. (2014), noting the large uncertainty and difficulty in assessing their accuracy. While we do not claim our model's estimates are more correct, its internal logic requires that Neanderthals maintain a high cultural carrying capacity and population growth rate to sustain a population of several thousand. This scenario implies they would have likely survived the HE4 event unless these rates collapsed suddenly, an outcome we consider improbable. Therefore, we argue that population size in the high thousands is unlikely, a perspective we have clarified in the revised text.

We conducted additional literature review:

Lalueza-Fox et al. 2005. Neandertal Evolutionary Genetics: Mitochondrial DNA Data from the Iberian Peninsula. Molecular Biology and Evolution. 22, 1077–1081.

Lalueza-Fox et al. (2005) analyzed mitochondrial DNA (mtDNA) from a ~43,000-year-old NEA from El Sidrón Cave (Asturias, Spain), finding that Iberian NEAs showed no significant genetic distinction from populations elsewhere. Their estimates of the female effective population size Nfe, ranging from 5,000 to 9,000, indicate that NEA genetic history was not shaped by a population bottleneck during the glacial maximum ~130,000 years ago. This aligns with our model conclusion that NEA populations recover from perturbations within 500–1000 years. Their Nfe estimates are similar to those for modern humans, suggesting similar demographic parameters — an approach we also adopted in our work.

To approximate the total NEA population Nc, we can apply standard population genetics conversions:

1. Accounting for Sex Ratio: Assuming an equal breeding sex ratio, the general effective population size is Nₑ ≈ 2 × Nfe = 10,000 – 18,000.

2. Accounting for Population Structure: Applying a common conversion factor of ~3 gives a total census size of Nc ≈ Nₑ × 3 = 30,000 – 54,000 individuals.

This figure likely represents a long-term average for the entire Pan-European/West Asian metapopulation, not the specific population of Iberia. The actual number would have fluctuated with climatic conditions, and the conversion factor, remains a subject of debate, with studies using values from 2 to 4.

The area of Iberian Peninsula covers about 582,000 km². A common estimate of the core range of the NEAs (primarily Europe up to 55°N and parts of Southwest Asia) is ~10 to 12 × 106 km². This implies that the Iberian Peninsula constituted roughly 5% of the total Pan-European/West Asian range of the Neanderthals. We can make a rough, area-based estimate for the Iberian population.

Iberian NEA Population ≈ Nc × 0.05 = 1,500 ~ 2,700

Bocquet-Appel, J.-P., Degioanni, A., 2013. Neanderthal Demographic Estimates. Current Anthropology. 54, S202–S213.

This article synthesizes multiple lines of evidence to estimate the Neanderthal metapopulation size and explain their cultural trajectory. It concludes that Neanderthals had a very small effective population size Nₑ, leading to a census size Nc estimated between 5,000 and 70,000 individuals. The key argument is that this chronically small and fragmented population, subjected to frequent climate-driven bottlenecks, created a "Boserupian trap." This means their low numbers limited their potential for technological innovation and cultural complexity, which may have been a decisive factor in their ultimate fate compared to modern humans.

Again, considering the area of the Iberian Peninsula is about 5% of the core range of the NEAs‘ territory, we have

Iberian NEA Population ≈ Nc × 0.05 = 250 ~ 3500

Prüfer et al. 2014. The complete genome sequence of a Neanderthal from the Altai Mountains. Nature. 505, 43–49.

Based on the genome from the Altai Neanderthal, they provide genetic evidence that the NEA metapopulation across its entire Eurasian range was consistently small. This conclusion is drawn from the genome's low genetic diversity, even lower than that of modern humans outside Africa, and signs of frequent inbreeding, such as the finding that the individual's parents were close relatives. This study infers a long-term small effective population size for the entire species, establishing that NEAs were a demographically fragile group. The study does not provide a precise numerical estimate but lays the groundwork for studies that calculated the female effective population size, pointing to a total census size in the tens of thousands.

Their Fig. 4 appears relevant to our study but somewhat difficult to interpret. It appears to have used the Pairwise Sequentially Markovian Coalescent (PSMC) model applied to the Altai Neanderthal genome. PSMC uses the pattern of heterozygosity (differences between the two copies of a ch

---

## [Decision Letter · Decision Letter 1]

2 Dec 2025

Pathways at the Iberian Crossroads: Dynamic

Modeling of the Middle–Upper Paleolithic Transition

PONE-D-25-33066R1

Dear Dr. Shao,

We’re pleased to inform you that your manuscript has been judged scientifically suitable for publication and will be formally accepted for publication once it meets all outstanding technical requirements.

Kind regards,

Marco Peresani

Academic Editor

PLOS ONE

Additional Editor Comments (optional):

Reviewers' comments:

Reviewer's Responses to Questions

**Comments to the Author**

1. If the authors have adequately addressed your comments raised in a previous round of review and you feel that this manuscript is now acceptable for publication, you may indicate that here to bypass the “Comments to the Author” section, enter your conflict of interest statement in the “Confidential to Editor” section, and submit your "Accept" recommendation.

Reviewer #1: All comments have been addressed

2. Is the manuscript technically sound, and do the data support the conclusions?

Reviewer #1: (No Response)

3. Has the statistical analysis been performed appropriately and rigorously?

Reviewer #1: (No Response)

4. Have the authors made all data underlying the findings in their manuscript fully available?

Reviewer #1: (No Response)

5. Is the manuscript presented in an intelligible fashion and written in standard English?

Reviewer #1: (No Response)

6. Review Comments to the Author

Reviewer #1: Thank you to the authors for taking the time to address all the comments in detail. The addition of new sensitivity analyses as well as specific reference to the results of all the sensitivity testing in the manuscript clearly demonstrate the robustness of the results. This allows the authors to offer interesting new hypotheses about Neanderthal and modern human population distributions and dispersal in Iberia that can be further tested with archaeological data.

7. PLOS authors have the option to publish the peer review history of their article (what does this mean?). If published, this will include your full peer review and any attached files.

Reviewer #1: No

---

## [Editor Report · Acceptance letter]

PONE-D-25-33066R1

PLOS One

Dear Dr. Shao,

I'm pleased to inform you that your manuscript has been deemed suitable for publication in PLOS One. Congratulations! Your manuscript is now being handed over to our production team.

Kind regards,

on behalf of

Dr. Marco Peresani

Academic Editor

PLOS One